# Convection-permitting regional climate simulations for representing floods in small and medium sized catchments in the Eastern Alps

Christian Reszler[1], Matthew Blaise Switanek[2], Heimo Truhetz[2]

[1]JR-AquaConSol GmbH, Steyrergasse 21, 8010 Graz, Austria
[2]Wegener Center for Climate and Global Change, University of Graz, 8010 Graz, Austria

*Correspondence to*: Christian Reszler. E-mail:  christian.reszler@jr-aquaconsol.at

**Abstract.** Small scale floods are a consequence of high precipitation rates in small areas that can occur along frontal activity and convective storms. This situation is expected to become more severe due to a warming climate, when single precipitation events resulting from deep convection become more intense ("Super Clausius-Clapeyron effect"). Regional climate model

(RCM) evaluations and inter-comparisons have shown that there is evidence that an increase in RCM resolution and in particular, at the convection permitting scale, will lead to a better representation of the spatial and temporal characteristics of heavy precipitation at small and medium scales. In this paper, the benefit of grid size reduction and bias correction in climate models are evaluated in their ability to properly represent flood generation in small and medium sized catchments. The climate models are sequentially coupled with a distributed hydrological model. The study area is the Eastern Alps, where

small scale storms often occur along with heterogeneous rainfall distributions leading to a very local flash flood generation. The work is carried out in a small multi-model framework using two different RCMs (CCLM and WRF) in different grid sizes. Bias correction is performed by the use of the novel Scaled Distribution Mapping (SDM, similar to usual quantile mapping) method. The results show that in the investigated RCM ensemble, no clear added value of the usage of convection permitting RCMs for the purpose of flood modelling can be found. This is based on the fact that flood events are the

consequence of an interplay between the total precipitation amount per event and the temporal distribution of rainfall intensities on a sub-daily scale. The RCM ensemble either lacks on one and/or the other. In the small catchment (< 100 km²), a favourable superposition of the errors leads to seemingly good CCLM 3km results both for flood statistics and seasonal occurrence. This is however, not systematic across the catchments. The applied bias correction only corrects total event rainfall amounts in an attempt to reduce systematic errors on seasonal basis. It does not account for errors in the temporal

dynamics and deteriorates the results in the small catchment. Therefore it cannot be recommended for flood modelling.

## 1 Introduction

Floods in small and medium sized catchments are often triggered by atmospheric processes on small scales, i.e., small scale frontal systems (Schemm et al., 2016) and convective storms. In the Austrian Alpine area, these types of small scale storms cause millions of Euros in damage every year. This situation is expected to become more severe as a result of a warming

climate and the Clausius-Clapeyron relationship. Single precipitation events are expected to become more intense (e.g., Allen and Ingram, 2002; Trenberth et al., 2003; Allan and Soden, 2008; Gobiet et al., 2013), and recent investigations have shown increases in deep convective precipitation can exceed the Clausius-Clapeyron relationship (known as the Super Clausius-Clapeyron Scaling effect, e.g., Lenderink and Van Meijgaard, 2009; Berg et al., 2013; Wang et al., 2017; Lenderink

et al., 2017).

Regional climate models (RCMs) are valuable tools for studying climate change effects on water resources. They are employed to generate climate simulations at scales below a 50 km horizontal resolution, like in the EU-FP7 project ENSEMBLES (Hewitt and Griggs, 2004) or the North American Regional Climate Change Assessment Program (NARCCAP) (Mearns et al., 2009). RCMs operating with 0.11° (~12 km) grid spacing became the standard in Europe as a

result of EURO-CORDEX (www.euro-cordex.net) (Jacob et al., 2014), which is the on-going European branch of the global Coordinated Regional Downscaling Experiment (CORDEX) (Giorgi et al., 2009) of the World Climate Research Programme (WCRP). Prein et al. (2016) investigated the added value in precipitation in the EURO-CORDEX RCMs. They demonstrated that as model resolution increased, atmospheric processes such as extreme precipitation are more realistically represented, especially in regions of complex terrain (e.g., the Alpine region). Nissen and Ulbrich (2017) focused on the representation of

heavy precipitation events in the EURO-CORDEX ensemble. They found that the frequency and size of heavy precipitation events are predicted to increase over most of Europe with increasing greenhouse gas concentrations. Moreover, the most severe events were detected to be in the projection period.

With improvements in numerical weather prediction (NWP) and computing technology, RCM grid spacing can now be further reduced to allow convection permitting climate simulations (CPCSs). CPCSs benefit from two major advantages with

respect to  precipitation extremes: (1) deep moist convection, which is the most important process in the majority of extreme precipitation events, are physically resolved by the RCM and (2) the representation of orography and surface fields is improved. Multiple studies have already demonstrated the added value of convection-permitting models (CPMs, Prein et al., 2015) in capturing extreme precipitation (e.g., Chan et al. 2013; Chan et al., 2014; Meredith et al., 2015; Chan et al., 2017; Zittis et al., 2017) and their frequency of occurrence (Ban et al. 2014; Knist et al. 2018). However, there are only a few

future projections that use CPCSs, like Prein et al. (2017), Ban et al. (2015), Kendon et al. (2014), and Knist et al. (2018). Although processes are better represented in CPCSs, it should be noted that local biases are not necessarily being reduced. Their bandwidths are large and (spatial and temporal) correlation coefficients are poor, when they are compared to highly resolved observation data (e.g. Prein et al., 2013; Ban et al. 2014; Knist et al. 2018). Especially, Ban et al. (2014) and Knist et al. (2018) found in common, that their models (CCLM and WRF) increasingly overestimate extreme events in

mountainous regions. This makes bias correction techniques indispensable, even if deep convection becomes resolved by RCMs. Also, additional computational costs are high which can limit an application particular for climate change studies in decision making.

Hinging on the scale of the driving data, climate change impact studies have often focussed on water balance in relatively large catchments (e.g., Fowler et al., 2007). Regarding floods, numerous studies were performed and pointed out the high

uncertainties in the GCM-RCM-hydrological model chain (e.g., Hennegriff et al., 2006; Dankers et al, 2007; Hanel and Buishand, 2010). Maraun et al. (2010) provided a comprehensive review on the requirements of hydrological models and their fulfilment via RCMs. They define the requirements in a correct representation of (1) intensities, (2) temporal variability, (3) spatial variability, and (4) consistency between different local-scale variables. Köplin et al. (2014) used future climate change scenarios from the ENSEMBLES project to analyse the seasonality and magnitude of floods in Switzerland. They found that the simulated change in flood seasonality is a function of the change in flow regime type. Magnitudes of both mean annual floods and maximum floods (in a 22-year period) are expected to increase in the future because of changes in flood-generating processes and scaled extreme precipitation. Using the new EURO-CORDEX models Alfieri et al. (2015) assessed projected changes in flood hazard in Europe based on the RCP8.5 scenario and the hydrological LISFLOOD model. Their results indicate that the change in frequency of discharge extremes is likely to have a larger impact on the overall flood hazard as compared to the change in their magnitude. On average, in Europe, flood peaks with return periods above 100 years are projected to double in frequency within 3 decades. In an effort to sequentially couple convection permitting RCMs with a hydrological model, first attempts have been made. For example, Kay et al. (2015) use results of 1.5 km RCM nested in a 12 km RCM driven by European-Reanalysis boundary conditions to drive a gridded hydrological model. However, they found that the 1.5 km RCM generally performs worse than the 12 km RCM for simulating river flows in 32 example catchments.

In this study, two regional climate models (CCLM, WRF) with different grid spacing (~50 km, ~12.5 km, and ~3 km) are sequentially coupled (one way) with a hydrological model for representing floods on small and medium spatial scales (30 km² to 1000 km²). An improved bias correction technique (Switanek et al., 2017) is used to minimize error propagation throughout the modelling chain. The study area is located in South-Eastern Austria (Styria), where local flash floods are the predominant flood type (e.g., Merz and Blöschl, 2003). The spatially distributed hydrological model KAMPUS (Blöschl et al., 2008) is used, which is in operational use for flood forecasting in Austria in small to medium sized scales (Blöschl et al., 2008; Ruch et al., 2012). The added value of the highly resolved convection permitting RCM setup (~3 km grid spacing) is evaluated in the period 1989-2010 by quantitative and qualitative criteria regarding flood generation.

**2 Study area and observation data**

The study area is located in South-Eastern Austria, at the border of the Eastern Alps (Fig. 1). Meteorological data of all available stations in the region were acquired from the Hydrographic Service of the provincial government of Styria and the Austrian Central Department of Meteorology and Geodynamics (ZAMG). Figure 1 shows the distribution of the stations between the period 2000 to 2009, which corresponds to the calibration period of the hydrological model. Data coverage has improved through the years by installing new stations. Historically in Austria, network of stations with daily data ("Ombrometer") is much denser than the network of stations with high temporal recording (e.g., every 15 minutes or hourly). In the bottom right plot the development of the station availability in Southern Styria is shown. At the beginning of 2000 the

number of stations with high temporal resolution significantly increased, whereas the number of stations with daily data was high since the beginning of the study period in 1989.

Interpolated fields of precipitation and air temperature are generated on an hourly basis. Stations with daily data are incorporated into the interpolation procedure to benefit from the dense network as follows (Reszler et al., 2006): First, daily data are interpolated on the model grid (1 km). Then, hourly data are interpolated on the same grid and the daily sum of the cells is calculated and scaled to the daily grid. Spatial distribution of daily precipitation is expected to be accurate even in the years before 2000, which is important for an accurate representation of the general water balance. However, due to the high spatial variability of precipitation in the region, hourly fields before 2000 contain more uncertainty. In contrast, uncertainty in interpolated hourly air temperature is generally much lower. The data were interpolated by a regression with station altitude and an interpolation of the residuals on the 1 km working grid. As an interpolation method for both variables the "Inverse Squared Distance" method was used. The interpolated fields for model calibration serve also as a reference data set for the RCM evaluation.

Runoff data for a high number of stream gauges are available at an hourly time step. These gauges are all used for model calibration (black triangles in Fig. 1, data provided by the Hydrographic Service of Styria). Representative gauges were selected in this study (labelled triangles with corresponding catchment boundaries, Table 1) in order to cover a wide range of catchment sizes (75 km² to 1100 km²) and different characteristics of soils and geology. There are more gauges used in the Western part, because the catchments vary largely by slope, climate, geology and soil type. This leads to differences in flood response and occurrence. For example, catchments of the gauges Schwanberg (S) and Voitsberg (V) reach relative high altitudes up to 2100 m a.s.l. at the Koralpe massif and are therefore expected to show significant influences of snow in winter and spring. Geology is crystalline (predominating gneiss and schist) with deep weathering zone (Flügel and Neubauer, 1984; BMLFUW, 2007) which implies significant storage capacities. Areas at the foot of the Koralpe consist mainly of tertiary sediments with low storage capacities (BMLFUW, 2007). Runoff at the corresponding gauges (e.g., gauge Gündorf – Gü) shows a relatively rapid response to rainfall and low baseflow (Ruch et al., 2012). The gauges Tillmitsch (T) covers the whole Lassnitz branch which flows into the Sulm which is gauged at the catchment outlet in Leibnitz (L).

In the Eastern part (so-called Grabenland creeks) only one gauge (Fluttendorf – Fl) is selected because of the relatively homogeneous climate, geology and soils. The runoff record extends over the whole simulation period and data are assumed very reliable according to the data provider Hydrographic Service. Geology and soils mainly consist of tertiary material. Influence of continental climate is increasing towards the east with values of annual precipitation in the order of annual evapotranspiration: Mean annual precipitation (MAP) is 700 mm in the East, whereas in the Western part, MAP ranges from 1100 mm at the foot to 1500 at the high altitudes.

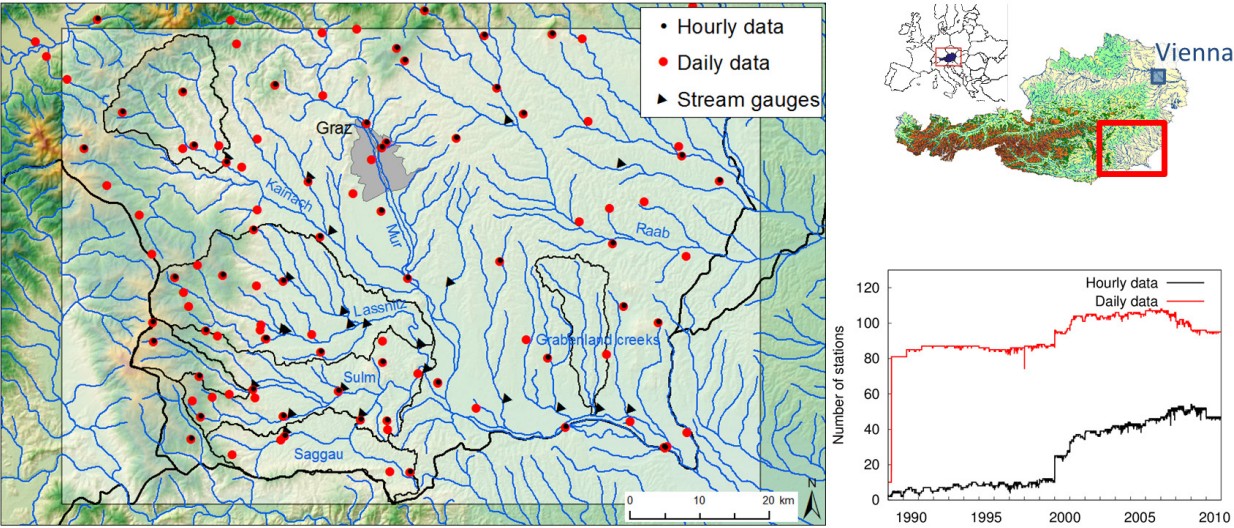

**Figure 1: Study area and station distribution (left) as well as availability of high resolution meteorological data (bottom right). Polygons are the catchment boundaries of the stream gauges for evaluation in this study (Table 1, Fig. 3).**

## 3 Method

### 3.1 Regional climate models

The RCMs we employ are the non-hydrostatic COnsortium for Small-scale Modeling (COSMO) model in CLimate Mode (COSMO-CLM or CCLM) (Böhm et al., 2006; Rockel et al., 2009) version 4.8 clm 17 and the Advanced Research version of the Weather Research and Forecasting model (WRF/ARW) (Skamarock et al., 2007) version 3.3.1. Both models are driven by the re-analysis dataset ERA-Interim (Dee et al., 2011) and cover the period 1989 to 2010. The models' innermost domain, the Greater Alpine region (GAR) with 3 km grid spacing, is reached via intermediate pan-European domains (without nudging) with 12.5 km grid spacing for CCLM and 50 km and 12.5 km grid spacing for WRF. By doing so, we mimic a typical setup as it is used in regional climate modelling applications and we do not run the risk of underestimating internal variability in our investigations. The simulations of the pan-European domains have contributed to the EURO-CORDEX initiative and have been evaluated in several studies, e.g. Katragkou et al. (2015), Kotlarski et al. (2014) and Prein et al. (2016). The model configurations for the convection-permitting (3 km grid spacing) simulations in the GAR are based on experiences from previous sensitivity experiments (Suklitsch et al. 2011; Awan et al. 2011; Prein et al. 2013; Prein et al. 2015). Our RCMs differ from their coarser resolved counterparts (EURO-CORDEX) insofar that the parametrization for deep-convection, the Tiedke scheme (Tiedke, 1989) in CCLM and the Kain-Fritsch scheme (Kain, 2004) in WRF, has been turned off in the GAR. Overview of the model domains and simulations used are given in Fig.2 and Table 2, respectively.

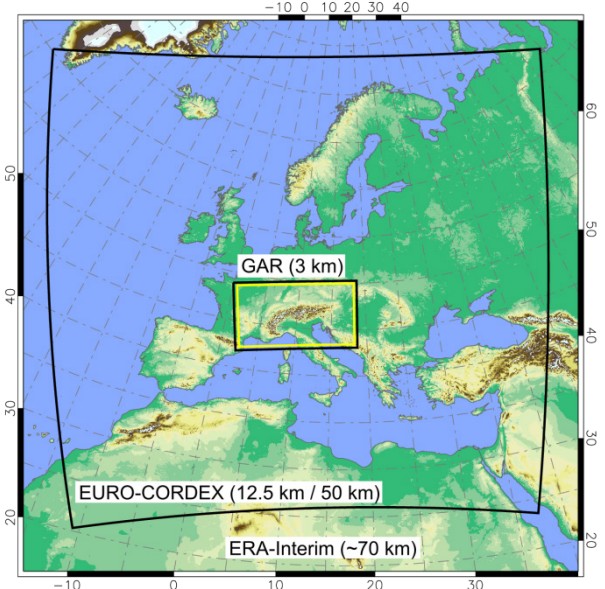

**Figure 2: RCM domains. ERA-Interim is dynamically downscaled with CCLM and WRF from its initial resolution of ~70 km to 3 km in the Greater Alpine Region (GAR) by making use of an intermediate model domain, the EURO-CORDEX domain with 12.5 km and 50 km grid spacing.**

**3.2 Error correction**

The novel method Scaled distribution mapping (SDM) is used to bias correct the model precipitation and temperature data time series (Switanek et al., 2017). SDM is a parametric method, but it is nearly identical to that of quantile mapping (QM) when correcting the historical period. However, for a "future" period (or any period outside of calibration), the method scales the observed distribution by the relative (for precipitation) or absolute (for temperature) distances between the

"future" and historical modelled CDFs. The commonly used bias correction method of QM (Wood et al., 2004; Piani et al., 2010; Themeßl et al., 2011; Teutschbein and Seibert, 2013) assumes that error correction functions can be treated as stationary from one time period to another. This assumption is responsible for altering the projected climate change signal. For example, a projected mean increase of precipitation of 20% can be inflated to be 30%, while extremes can be altered even more dramatically. However, Maraun (2012), Teutschbein and Seibert (2013), Maurer and Pierce (2014) and Switanek

et al. (2017) showed this assumption of a stationary error correction function to be invalid, and as a result, the altering of the raw model projected changes to precipitation and temperature was found to be unjustified. In addition, quantile mapping was found to overestimate values of low precipitation and underestimate high precipitation (Maraun, 2013). SDM, in contrast, does not rely on a stationary error correction function, but rather attempts to best preserve the raw model projected changes across the entire distribution. However, the over (under) estimation of low (high) precipitation intensities remains. Bias

correction was performed on RCM precipitation and temperature data independently for each grid cell and calendar month. It was implemented on a 3-hourly window to more accurately capture the observed diurnal cycle.

### 3.3 Hydrological model

The spatially distributed model KAMPUS (Blöschl et al., 2008) is used, which is in operational use for flood forecasting in Austria. It contains conceptual models for snow melt, soil moisture accounting and flow routing. The snow model is based on the degree-day approach which calculates snow melt depending on the air temperature. For snow accumulation precipitation is split into snow and rainfall by a lower and an upper threshold temperature with a linear transition. Depending on the actual soil moisture, rainfall and snowmelt are non-linearly partitioned into a component that increases soil moisture and a component that contributes to runoff, dQ. Soil moisture can only be depleted by evapotranspiration. Runoff routing on a raster cell (hillslope) is represented by an upper zone and two lower zones, which are formulated as linear reservoirs. dQ is the input into the upper zone. The zone has three outlets: (i) outflow with a low storage coefficient ($k_1$) that represents interflow, (ii) percolation to the lower reservoirs (saturated zone), and (iii), when a defined threshold, $L_1$, is exceeded, outflow with a very low storage coefficient ($k_0$) representing surface or near surface runoff. The percolation rate into the two lower zones is separated into two components by a factor. Outflow of the lower zones is defined as groundwater flow and deep groundwater flow, respectively. A bypass flow, $dQ_{by}$, routes rainfall and snow melt directly into the lower storages (macro-pore flow). Model structure is described in detail in Blöschl et al. (2008). In this work, the original vertical structure is extended by a module for infiltration excess. At very high intensities ($I > I_{crit}$) parameters of soil storage are reduced, and bypass and deep percolation is set to zero. Values for $I_{crit}$ and the reduction of infiltration parameters are obtained by calibration.

Total runoff on a grid cell is calculated as the sum of the outflows from all zones. It is then aggregated to sub-catchments and convoluted by a linear storage cascade which represents runoff routing in the stream network within each of the sub-catchments. Routing in the river reaches which connect model nodes is formulated by a cascade of linear reservoirs (Reszler et al., 2008b). By a step-wise linear formulation, this model allows for incorporating non-linear effects in flood rooting, such as flood wave acceleration at high water levels and flood retention at flood plains. For the latter, board-full discharge and existing 2D hydrodynamic studies have been provided by the Hydrographic Service of Styria for calibrating the corresponding parameters. This is particularly important for a plausible representation of flood peak attenuation at very large floods. Since the hydrological model is also driven by simulated, often biased, precipitation input, flood peaks may be simulated which exceed observations.

The model domain extends over all of Southern Styria (grey shaded window in Fig. 1). The Western part has previously been calibrated (Ruch et al., 2012), as it is implemented for operational flood warning by the provincial government of Styria. The Eastern part was extended in the current study. The model has a sub-catchment structure with 96 catchments and 152 internal model nodes (Fig. 3). The model is driven by precipitation and air temperature with an hourly temporal resolution and a 1 km gridded spatial resolution. No further climate variables are required; the potential evaporation is represented by the modified Blaney-Criddle method (Schrödter et al., 1985), which only requires air temperature as input.

The method of extending the model to the Eastern domain followed the strategy outlined by Reszler et al. (2006, 2008a). This approach contains several steps for parameter identification based on the Dominant Processes Concept (e.g., Grayson and Blöschl, 2002), and proposes the usage of auxiliary information and data (e.g., field surveys, snow depths, hydrogeological data) and the stratification into different event types (convective, advective and snow melt events). Spatially distributed information is incorporated in a GIS framework, but the resulting hydrotope structure is manually fine-tuned. The following hydrotope types were chosen (compare to Reszler et al., 2006): urban areas, low density urban areas, steep slopes open, steep slopes forest, flat agricultural areas with porous aquifer, saturation areas and karstic areas. Hydrotope structure and parameter values are chosen in consistency with the existing model in Western Styria, where in some catchments (e.g., at the foothills of the Koralpe massif) the physiographic situation is similar.

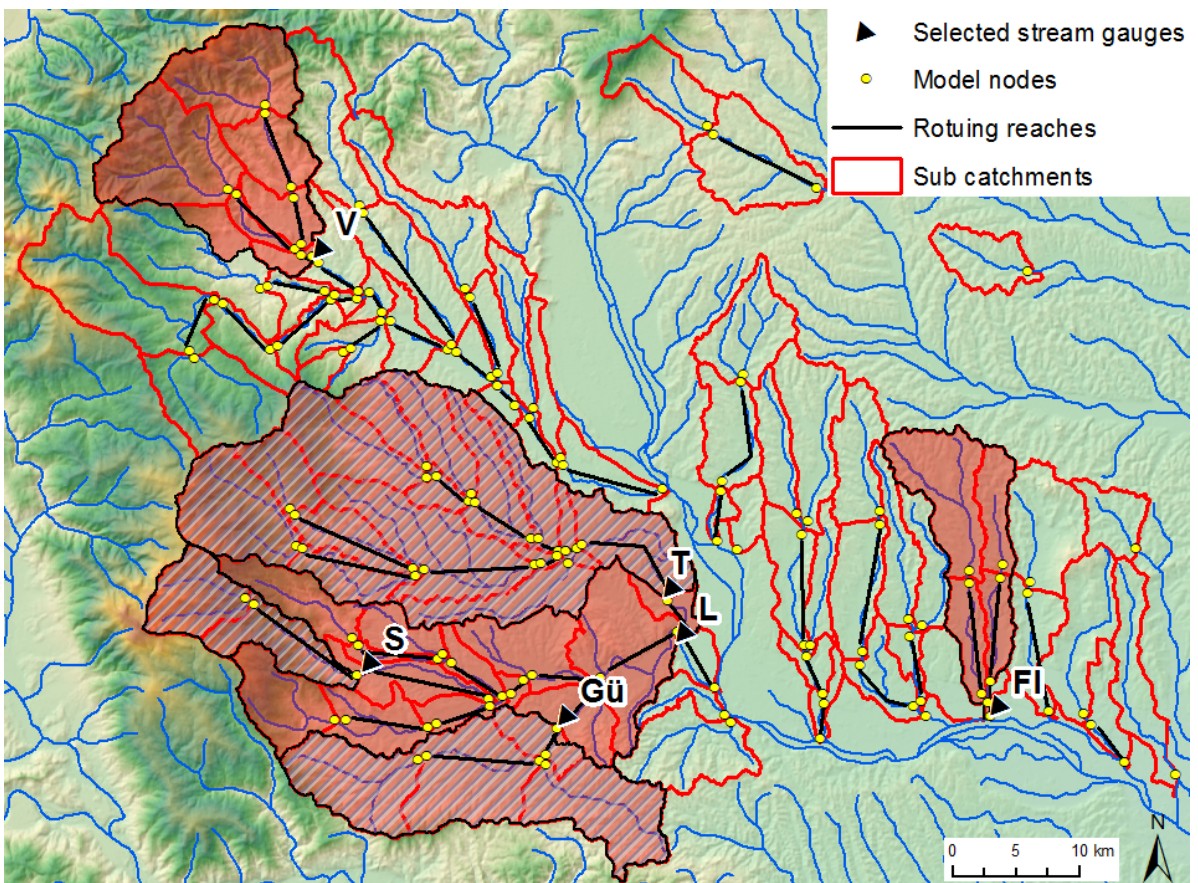

**Figure 3: Spatial model structure (sub-catchments, nodes, routing reaches), available gauges for calibration and catchments of stream gauges for evaluation highlighted (nested catchments are shaded). Evaluation gauges see Tab. 1.**

### 3.4 Evaluation measures

In order to combine quantitative and qualitative evaluation of the different model simulations, the following measures are chosen:

- Catchment size as an indicator for general attenuation affects
- Frequency of floods, i.e. maximum annual floods (MAF)
- Seasonality of floods
- Other variables, such as soil moisture (simulated by the hydrological model)
• Event-based analyses (performance at particular events, event/weather types)

Catchment size is implicitly incorporated by the selection of the gauges with a wide range of catchment areas from small to medium scale (75 to 200 km²) as well as the larger catchments of the gauges downstream (< 1100 km²) (Table 1). In the evaluation plots in this paper the size is identified by the letters "L" for large, "M" for medium and "S" for small. In

addition, differences between the catchments in runoff generation and response times are evaluated by different model parameters obtained by the calibration.

Frequency of floods are analysed by typical statistics of maximum annual flood peaks using the following "plotting position" (Weibull)

$$RP = \frac{(N+1)}{m} \tag{1}$$

*RP* is the (empirical) return period, *N* the number of values (years) and *m* the ranking (1 for the maximum and N for the minimum flood).

The seasonality of floods gives first insights into the main hydrological drivers for flood occurrence (Parajka et al., 2010). It is the result of the relative influences of soil moisture, evaporation and snow processes and varies considerably in space. In their event type analyses, Merz and Blöschl (2003) used the seasonality of maximum annual flood (MAF) peaks as an

indicator describing the timing of floods. Here, seasonality is first, analysed simply by counting MAF peaks in the four seasons December, January, February (DJF), March, April, May (MAM) June, July, August (JJA), and September, October, November (SON). Second, in order to illustrate seasonality for different simulation runs in the small multi-model ensemble, circular statistics are performed. For each event the date of occurrence of the MAF is transposed to an angle by

$$\alpha_i = D_i \frac{2\pi}{365} \qquad i = 1, \dots n \tag{2}$$

where $D_i$ denotes the day of the year ($D_i = 1$ for Jan. 1st, $D_i = 365$ for Dec. 31st). This angle is averaged by the following equations

$$Y = \frac{\sum_{i=1}^{n} \sin(\alpha_i)}{n}$$

$$X = \frac{\sum_{i=1}^{n} \cos(\alpha_i)}{n}$$

$$r = \sqrt{X^2 + Y^2} \tag{3-6}$$

$$\theta_r = \arctan\left(\frac{Y}{X}\right)$$

where X and Y are the rectangular coordinates of the mean angle $\Theta_r$, and $r$ is the mean vector length, which is a measure of strengths of the seasonality ($r = 1$ if all events occur at the same date). Note that the final resulting mean angle depends on the quadrant of the calculated mean angle.

Using a hydrological model for an evaluation of climate model results also enables the incorporation of other hydrological quantities, which give indications about the performance of the climate model regarding the hydrological conditions. Soil moisture is an important variable to be analysed in terms of non-linearity and threshold processes in flood generation (e.g., Penna et al., 2011). It is continuously calculated by the hydrological model, and hence, can be used as a comparison between the different simulation runs.

At last, mainly using the 3 km convection permitting RCM results, runoff simulations at characteristic events are checked for their realistic event evolution and the plausibility of the corresponding atmospheric and hydrological conditions.

## 4 Added value in RCMs due to increased resolution

In order to demonstrate added value due to a reduction in the model grid spacing, we derived averaged precipitation fields of the models and the observational data and calculate the spatial correlation coefficient between them. Figure 4 illustrates the resultant correlation coefficients for all models, months and hours of the day. Higher correlations for both models, illustrated by the warmer colours, are more clearly observed towards the left side of Fig. 4, the side where highest model resolutions are depicted. This shows that the RCMs improve, on average, in their ability to simulate precipitation fields across space as the resolution of the model increases.

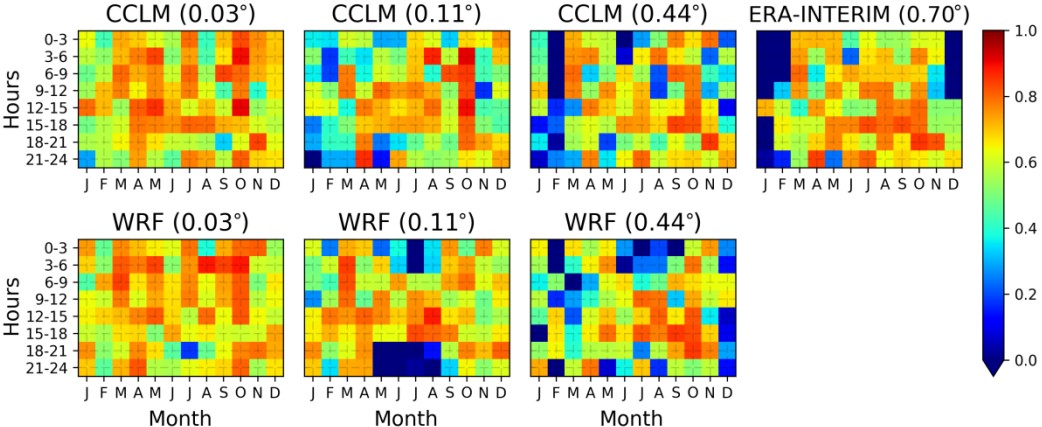

**Figure 4: Added value of using higher model resolution. The colour bar corresponds to the correlation coefficients between the observed and the modelled spatial fields of averaged precipitation. The x-axes and the y-axes show the months of the year and the hours of the day, respectively.**

Added value is also seen on catchment averaged quantities. Generally, the convection-permitting models increase precipitation intensities from heavy (>90[th] percentile) precipitation events in all catchments. In the case of CCLM, this

results in added value (together with some overestimation), since the coarser resolved counterpart CCLM 0.11° largely underestimates (in the range of -16% to -26% across the catchments) precipitation intensities on average (see Figure 5). In contrast, WRF does not show such strong underestimations in the 0.11° simulation and WRF 0.03° gives an overestimation, because of its linkage to the coarser resolved 0.11° simulation that enables error propagation (Addor et al., 2016). The reasons for the enhancement of intensities in mountainous regions may be a result of the higher resolved orography and is in agreement with previous evaluation studies (Knist et al., 2018; Prein et al., 2013; Ban et al., 2014; Langhans et al., 2013). Note, WRF 0.11° is generally in a better agreement with the observations than CCLM 0.11° (Figure 5), although both simulations are of comparable performance on the European domain (e.g., Katragkou et al. 2015; Kotlarski et al. 2014) and add value in mountainous regions compared to their 0.44° counterparts (Prein et al., 2016).

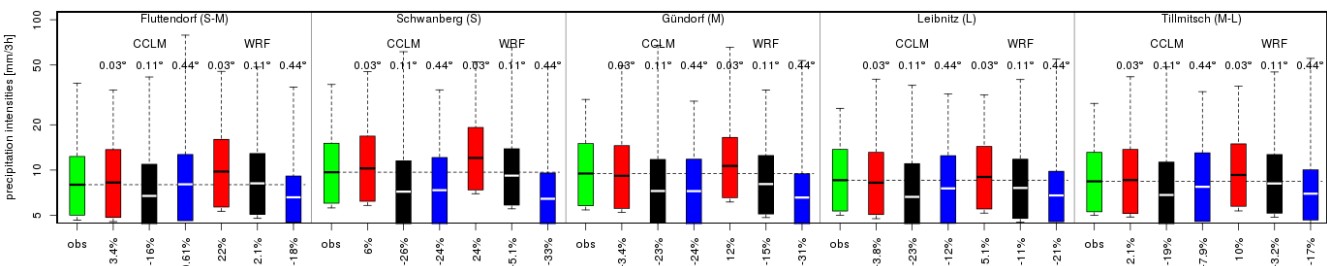

**Figure 5: Catchment averaged heavy (>90[th] percentile) precipitation intensities of observations (green) and CCLM and WRF with 0.03° (3 km; red), 0.11° (12.5 km; black), and 0.44° (50 km; blue) grid spacing. Y-axis has a logarithmic scale. Coloured boxes indicate the 10[th] – 90[th] inter-quantile range, horizontal markers in the boxes denote mean values, whiskers refer to maximum values. Relative biases of mean values are given along the x-axis.**

## 5 Hydrological model calibration and validation

The hydrological model was calibrated, for each sub-catchment, against runoff data of all available stream gauges in the period 2000-2009 (Fig. 1). Calibration results in Western Styria are available, and the found parameters in catchments with similar soil and geological properties serve as a priori values for the catchments in the extended part. The historical data in the current study (1989-1999) are used for model validation. This allows also a validation of the existing model; these data were provided for the current study and had not been used for model calibration. Quantitative metrics such as the commonly used Nash-Sutcliffe-Efficiency (NSE, Nash and Sutcliffe 1970), the BIAS based on mean runoff values, and the Root Mean Square Error (RMSE) are used to measure model calibration. In Table 3 the results for the selected gauges are listed. As it is often the case, NSE is lowest in the smaller catchments, e.g., Schwanberg and Fluttendorf with 0.77 and 0.78 in the calibration period, respectively. In the validation period the NSE falls below 0.7 in these two catchments. The historical period also includes phases with poor data availability (see Fig. 1), which is also the reason for the drop of the NSE value in the validation period at the Gündorf gauge.

Examples of hydrographs in the calibration and validation period are attached in supplementary material (Fig. S1 and Fig. S2). In addition to flood peaks, runoff generation and rainfall response is represented very well. Differences in the shape of

hydrographs are also accurately simulated. For example, the Schwanberg gauge shows short peaks due to short concentration times in the small catchment but at the same time high baseflow. The latter indicates high fraction of slowly draining flow component (groundwater) from long term storage. On the other hand, in the medium sized catchment Gündorf, short peaks also indicate short response times, but baseflow is significantly lower. This difference can be attributed to the different geologic conditions in the area. In the Schwanberg catchment the significant subsurface storage can be attributed to a deep weathering zone overlaying schists and gneiss, and geology in the Gündorf catchment consists mainly of tertiary material (silt, loam) with very low storage capacities. In the larger catchments flood peaks are smooth lasting over several hours which shows the attenuation effects (Tillmitsch, Leibnitz). The resulting model parameter values representing time scales of runoff response show the flashy character in the catchments: The time constant of the fast flow component, i.e. surface runoff in open steep slopes ($k_0$), at hillslope scale is in the order of simulation time step of 1 hour, routing within the sub-catchments (30 - 50 km²) is in the same order, and travel time within most river reaches connecting the nodes is 2-3 hours.

For this study, representation of flood frequency is important. Model simulated maximum annual floods for the entire study period (calibration and validation period combined, 1989-2010, 22 years) are compared with observed flood peaks in Fig. 6 (flood frequency plot). Although the MAF distribution was not explicitly subject to calibration, and the data availability was relatively poor in the period 1990-2000, the model accurately simulates observed flood statistics at the selected gauges. The largest flood is simulated well at all gauges, while simulation results at the smaller events are reasonable. Both in the calibration and validation period, deviations at significant events are analysed in terms of probable errors in input (precipitation), model structure, model parameters and/or runoff data. At the exceptional events threshold processes are operative, which are accurately simulated. For example, at the largest flood at the Schwanberg gauge in August 2005 (Figure 6, plot above left, extrapolated RP would be more than 100 years) was a very local event (see following Fig. 9 lowest panel) and the interpolated rainfall is assumed to be relative uncertain. In order to simulate the observed flood peak parameters would be needed which are not plausible and decrease model performance at other large events. Also inundation occurred during the event in the Schwanberg town, which likely led to uncertainties in the observed peak runoff data. At the Fluttendorf/Gnasbach gauge, flooding occurred at the two events in 2009 (see Figure 15), and retention by inundation in flood plains was calibrated successfully in the flood routing model. At the Voitsberg gauge the two largest floods were slightly underestimated. The largest flood occurred in the October 1993, within the validation period, which was underestimated in the simulation. The largest flood in the simulation is the 2009 flood which was represented very well (see supplementary material). Data quality used to be poor in 1993 in the high altitude catchment in the North-Western part. Station density in this part today is still lower than for example, in the South-Western part (see Fig. 1). Same situation can be stated for the Tillmitsch gauge. At this gauge, four medium event peaks (from 92 to 117 m³/s) are the MAFs in the years 1993 – 1996, and the simulated flood peaks at the corresponding plotting positions were slightly underestimated.

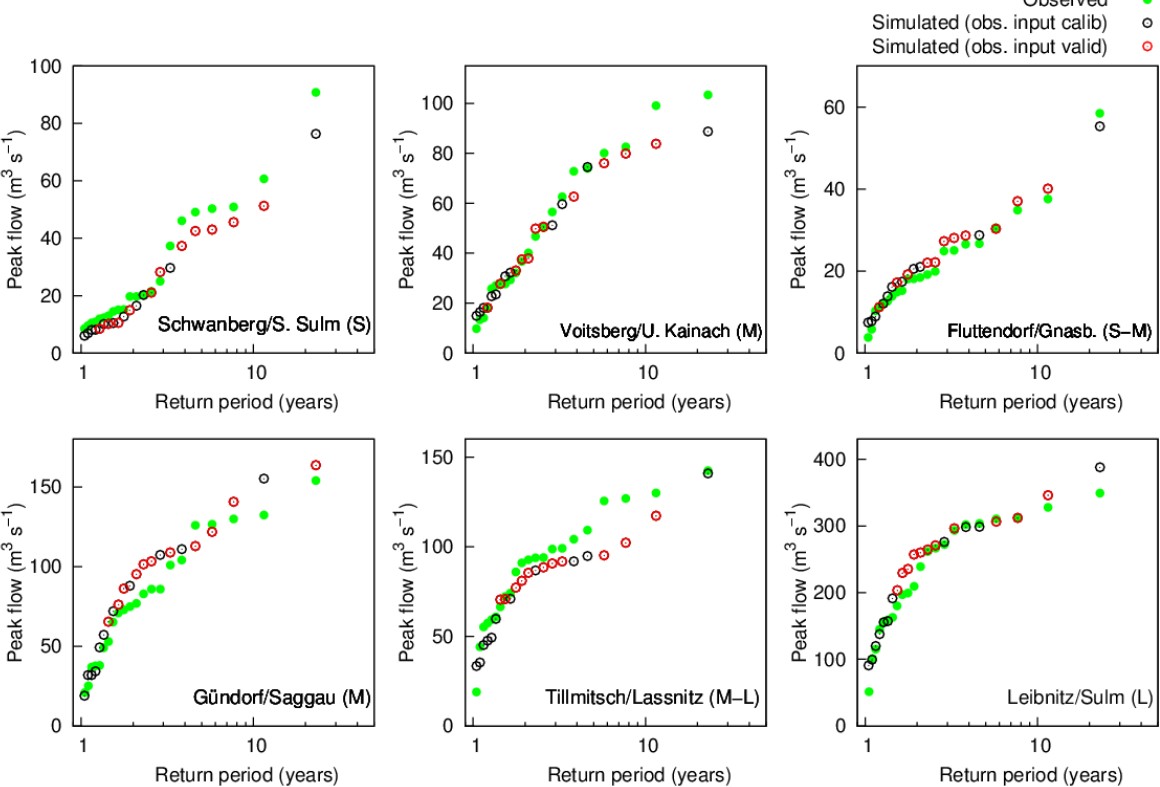

**Figure 6: Simulated and observed maximum annual flood peaks vs. empirical return periods (Eq. (1), flood frequency plots) of the selected gauges in the period 1989-2010. The peaks in the validation period are marked with red colour.**

## 6 Evaluation of simulation results using RCM data as input

5    With the calibrated hydrological model, simulations are performed using the results of the RCMs as input. In the following sections 6.1 and 6.2 evaluation results are discussed in detail for the runoff simulations using the CCLM results. The same procedure has been applied using the WRF results (provided in supplementary material). In a synthesis step (section 6.3), all the results of the small multi-model ensemble are summarized and compared for formulating final conclusions.

### 6.1 Uncorrected RCM data

10    Flood frequency plots for the selected gauges using uncorrected CCLM data (and the ERA-Interim data) as input compared to the observations are shown in Fig. 7. The figure illustrates the improvement of the results using the 3 km CCLM data, particularly for the smallest catchment Schwanberg/S. Sulm with a size of app. 75 km². In this specific region, the convection permitting simulation seems to be a necessity in order to accurately represent the magnitude of floods. In some larger catchments the simulations with the coarser RCM data already yield reasonable results. As the plotting positions

suggest, the statistical properties, mean and standard deviation, decrease with increasing grid size. The skewness does not decrease; in Voitsberg and Schwanberg the CCLM 0.11° simulation yields a high skewness. At the latter only three events are simulated with peak flows above 20 m³/s, whereas the observation reaches a maximum of 90 m³/s.

Most of the RCM settings show negative biases regarding MAF peaks; however, some are significantly positively biased, e.g., Voitsberg/Kainach in the North-Western Alpine part At the Fluttendorf gauge (upper right sub-plot) the 0.44° data lead to a maximum flood that significantly exceeds the observations; this peak is the consequence of overestimated heavy rainfall intensities (app. 300 mm in 18 hours), which was simulated in August 2005 during the "Alpenhochwasser." For comparison, the observed MAF at the Fluttendorf gauge occurred during an extreme precipitation event in June 2009 (Figure 15 in the following section).

The seasonal occurrence (winter: DJF, spring: MAM, summer: JJA and autumn: SON) of the simulated MAFs is analysed in Fig. 8. The improvements are evident when reducing grid size; the simulation with the uncorrected 3 km CCLM data represents the observed seasonality very well. The figure further shows that both the CCLM with 0.44° (~50 km) and 0.11° (~12.5 km) grid sizes yield a shift of the flood season from summer (JJA) to spring (MAM) in all catchments - except Schwanberg. Also, in the catchments in Western Styria (Kainach, Sulm, Saggau, Lassnitz) numbers of MAF in autumn are underestimated in all CCLM settings. In autumn, frequently occurring low pressure systems in the Mediterranean or east of the Alpine region induce heavy rainfall which can often lead to large floods (Seibert et al., 2007). The simulations indicate that this is underrepresented in all CCLM data. This shows the value of the use of seasonality for an evaluation of an accurate representation of the main flood generating mechanisms. Flood statistics in the mentioned cases yielded reasonable results, but this criterion alone could be misleading. In the catchment in Eastern Styria (Gnasbach) the relatively uniform distribution is captured well (upper right sub-plot). Both for flood frequency and seasonality, using the WRF data the results are worse than using the CCLM data (shown in supplementary material, Fig. S3 and Fig. S4, and discussed later in section 6.3).

Simulated soil moisture on monthly basis (attached in supplementary material, Fig. S5 above) shows annual dynamics that are similar to the seasonality of MAF. Also, the improvements using the 3 km CCLM (0.03°) compared to the coarser resolution are evident, and ERA-Interim is closest to the reference. Within ERA-Interim the observed situation is represented, however, the coarse resolution also leads to a bias. Underestimation in summer is significant, particularly in the case of the 0.44° (~50 km) and 0.11° (~12.5 km) grid sizes. In this season heavy storms occur with often convective character or double events. The corresponding flood magnitude is (non-linearly) controlled by the antecedent soil moisture, which is the consequence of the meteorological and hydrological history prior to the flood events. The same is true for the autumn (SON in Figure 8), when the soil moisture is underestimated and often floods occur as a consequence of Mediterranean low pressure systems in combination with wet soils due to reduced evapotranspiration.

As the first results show, the CCLM 3 km setting yields a clear benefit regarding magnitude and frequency of large floods particularly in small catchments. As stated above, the floods of the simulations are not necessarily aligned in time with observations. Figure 9 shows two simulation periods for the gauge Schwanberg/S. Sulm (75 km²). In the panel above the

period August and September 1996 is plotted, when the largest flood was simulated with the 3 km CCLM input. There are several small rainfall events in the observation but no large flood occurred during this period. The panel below shows the largest flood in the record which occurred in August 2005 ("Alpenhochwasser"). This flood was completely missed using the CCLM input. As it happens, the size and the month of occurrence of the two simulated floods is the same. Also, in 1996 the

temporal rainfall distribution simulated by the climate model, showing a very high one hour block embedded into a slight pre- and post-rainfall, leads to a plausible shape of the hydrograph. This example indicates that also for small catchments, large floods are "produced" which leads to a rather good statistical representation of maximum annual flood peaks (see Fig. 6), but an event-by-event comparison partly fails, because of the large RCM domain: the 3 km CCLM simulation is driven by a 12.5 km CCLM simulation covering the European domain that is in turn driven by re-analysis data (ERA-Interim)

without nudging. Due to the internal model variability, the 12.5 km simulation partly deviates from ERA-Interim and even synoptically forced events (like the 2005 flood) may not be correctly represented in space and time in the RCM. This decoupling-effect is well known in Regional Climate Modelling and Numerical Weather Prediction and was first published by Kida et al. (1991). Along the modelling chain, the convection-permitting 3 km simulation is affected by decoupling for two times: (1) via the 12.5 km domain that is partly decoupled on the synoptic scale and (2) via its own internal variability,

so that single thunderstorms (under weak synoptic forcing) may occur at different places or/and at different time as in the observations ("double penalty" problem). This limits the applicability of event-by-event comparisons and emphasises statistical evaluation approaches.

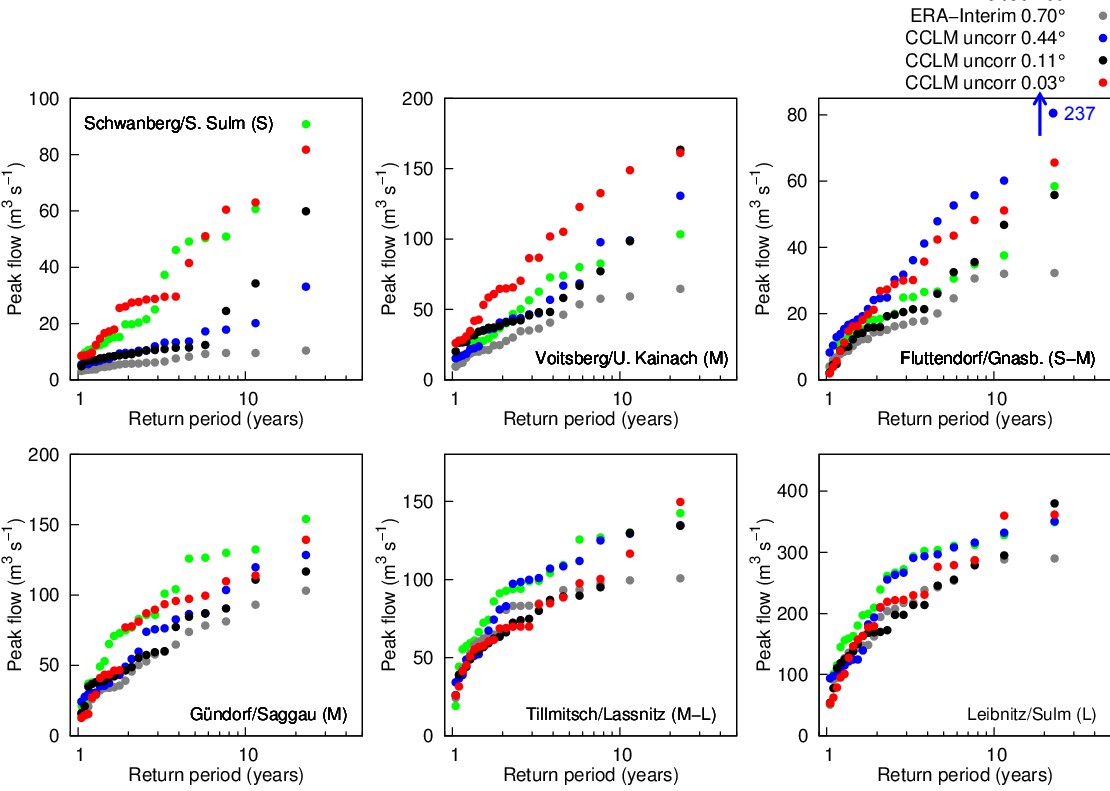

**Figure 7: Simulated maximum annual flood peaks using raw CCLM data as input and observed maximum annual flood peaks vs. empirical return periods (Eq. (1), flood frequency plots) of the selected gauges in the period 1989-2010.**

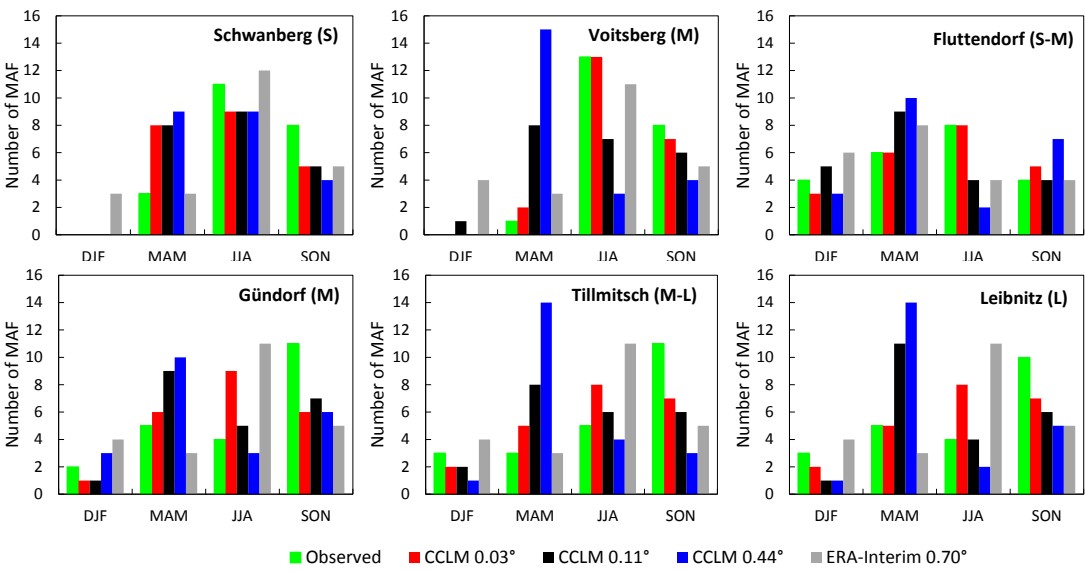

5   **Figure 8: Number of maximum annual floods in the four seasons (seasonality) from the simulation using raw CCLM data as input compared to the observation at the selected gauges in the period 1989-2010.**

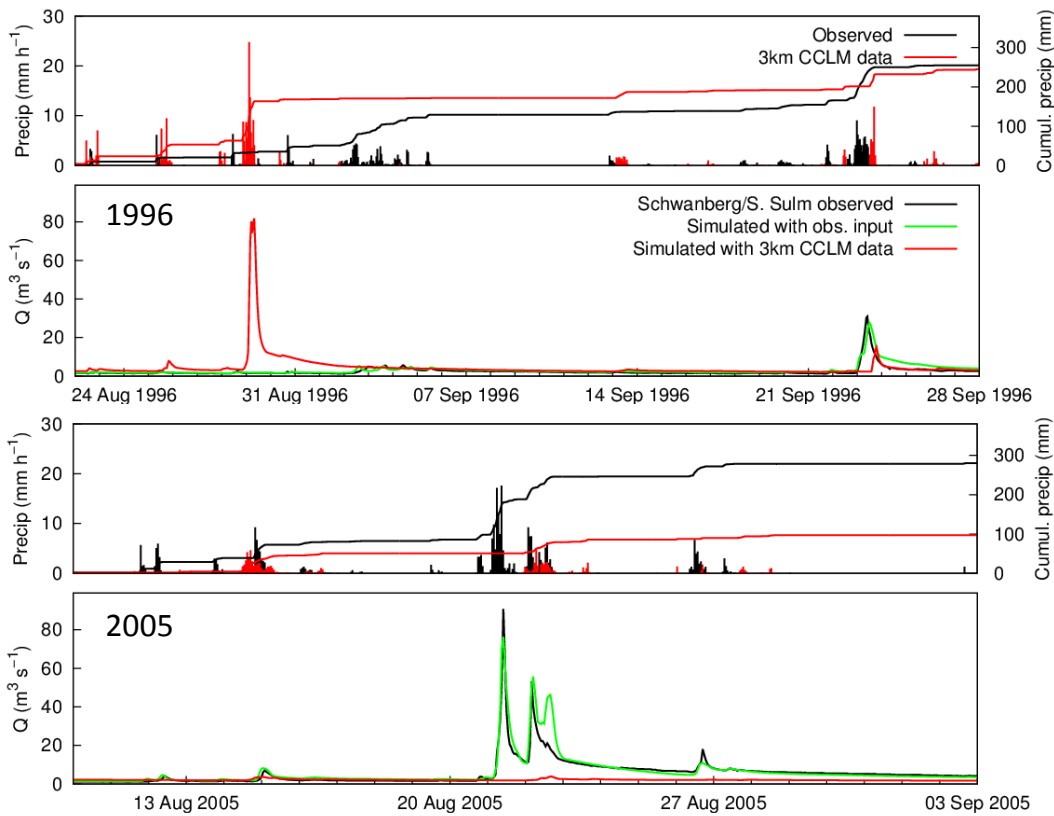

**Figure 9: Example of two events (Aug. 1996, above and Aug. 2005 below, in each case plotted with catchment precipitation above the runoff) simulated with raw CCLM 3 km data compared to the simulation with observed input and observed runoff data for the Schwanberg gauge.**

## 6.2 Bias corrected RCM data

In the same way as for the raw RCM data, the hydrological model is driven using bias corrected data. After bias correction results of flood statistics using CCLM (Figure 13) are improved, except for the smallest catchment Schwanberg. Here, particularly the results deteriorate compared to the run using the uncorrected data (Fig. 6).

This can be explained by an interference of the temporal distribution of precipitation intensities during the flood generating rainfall events and the bias correction that simply ignores such temporal relationships. Figure 10 shows the precipitation intensities that contribute to the maximum annual flood events in Schwanberg simulated by CCLM 3km, before and after bias correction. Each event is limited to a duration of two days before the maximum peak flow is reached. Figure 10a demonstrates the work of the bias correction that removes severe under (over) estimation of low (high) intensities in the CCLM 3km data, but leaves the total amount of precipitation of these events largely unaffected so that a median under catchment of -15% to -16% remains (Figure 10b). The success of CCLM 3km in capturing the flood events (Figure 13) lies in the precipitation amount that is accumulated over a shorter time period prior to the flood events. Figure 10c shows the averaged relative bias of accumulated precipitation as a function of the accumulation time prior to the event. On average,

CCLM 3km increasingly overestimates the accumulated precipitation as the accumulation time is shortened. The lack of total precipitation is compensated by the temporal evolution that gives about 20% more precipitation within a time range of 24 h before the flood event. The bias correction removes these (compensating) overestimated intensities and the positioning of the peak flows (Figure 13) rapidly drops. Note, the reason why single intensities are not properly corrected is based on the fact that the bias correction is independently applied on each grid cell. The remaining deviations from the observations (Figure 10a) result from the aggregation of single grid cells to areas that cover the entire catchment.

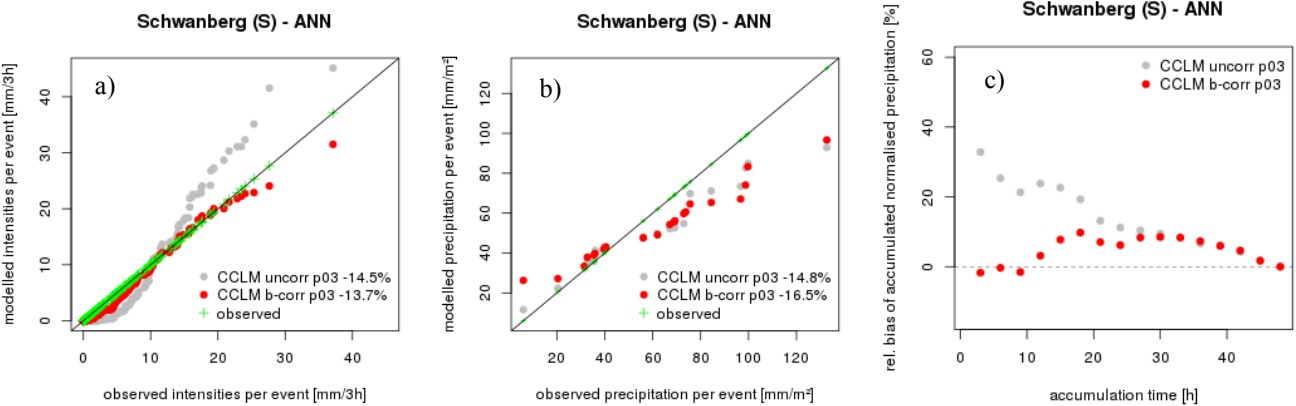

**Figure 10: Comparison between modelled (CCLM 3km), bias corrected, and observed precipitation characteristics tied to the 22 MAF events in the catchment Schwanberg. The left panel shows the single precipitation intensities. The middle panel depicts the total precipitation per event (defined as the 2 day period before the maximum peak flow). The right panel shows the (event averaged) relative bias of the accumulated precipitation amount (normalised by the events' total precipitation) prior to the flood events as a function of its accumulation time. Numbers in the legends give the relative median bias of the plotted data.**

In contrast, the positioning of WRF 3km peak flows in Schwanberg lies above the observations and the bias correction leads to a deterioration (Fig. S3). In this case, WRF 3km overestimates precipitation intensities across the flood events and the bias correction changes this (due to the aggregation of single grid cells to catchments) into an underestimation (Figure 11a). This leads to an overestimation (underestimation) of event-related precipitation amounts (Figure 11b) for the uncorrected (corrected) data. In WRF 3km the temporal distribution of the intensities is in a much better agreement with the observations than in CCLM 3km (compare Figure 11c and Figure 10c). However, since the total amount is overestimated, the peak flows are higher. The bias correction furtherly deteriorates the temporal distribution of the intensities that lie closer to the flood event and together with the underestimation of the total amount this gives a rapid drop in the positioning of the peak flows

(Fig. S3). Note, this good representation of the temporal distribution in WRF 3km is a catchment specific feature.

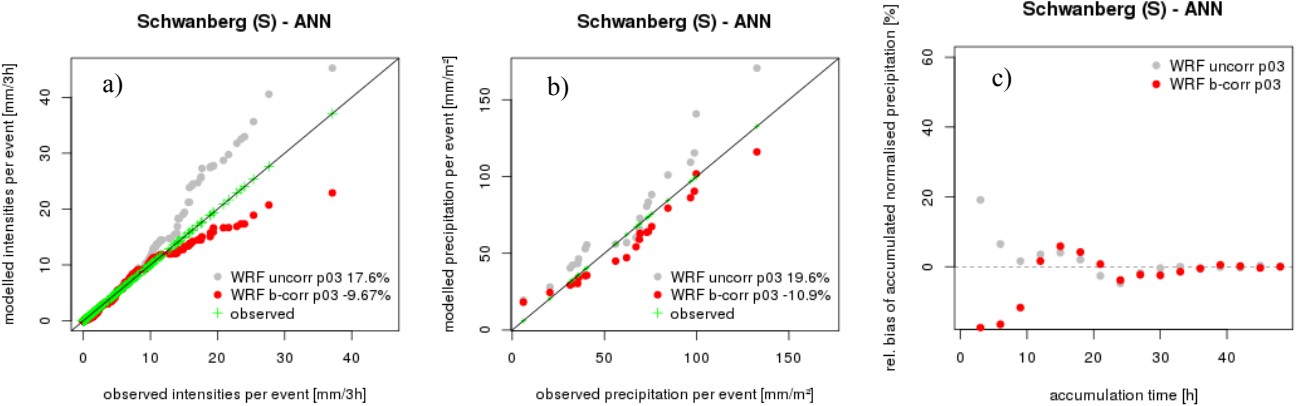

**Figure 11: Same as Figure 10, but for WRF 3km.**

Also, in the small catchments, the aggregation to 3hr sums has an influence on the performance. We tested it by using the 3hr sums of the CCLM 3km and comparing to the 1 hr results (not shown). There is a decrease of flood peaks, but the main decrease in performance in the small catchment Schwanberg is due to the error correction explained above.

In some cases bias correction leads to an over-compensation of the flood peaks, particularly in the case of the ERA-Interim data. For instance in Gündorf, flood event related precipitation intensities and amounts are largely underestimated in ERA-

Interim by more than -30% on average (median) (Figure 12ab), but the precipitation amount within an time range of ~3/4 day before the flood event is over estimated by ~25% on average (Figure 12c). However, this overestimation is too small and the peak flows of the corresponding flood events (Figure 13) stay below the observations. The bias correction overcorrects the catchment averaged intensities that are larger than ~7 mm/3h (Figure 12a) and leaves smaller intensities undercorrected (as an effect of catchment aggregation) which however yields to a well representation of the precipitation amounts (Figure

12b). The overcorrection of higher intensities leads to a further increase of the accumulated precipitation amount 3/4 day prior to the flood events and the corresponding positioning of the peak flows (Figure 13) lie above the observations in general.

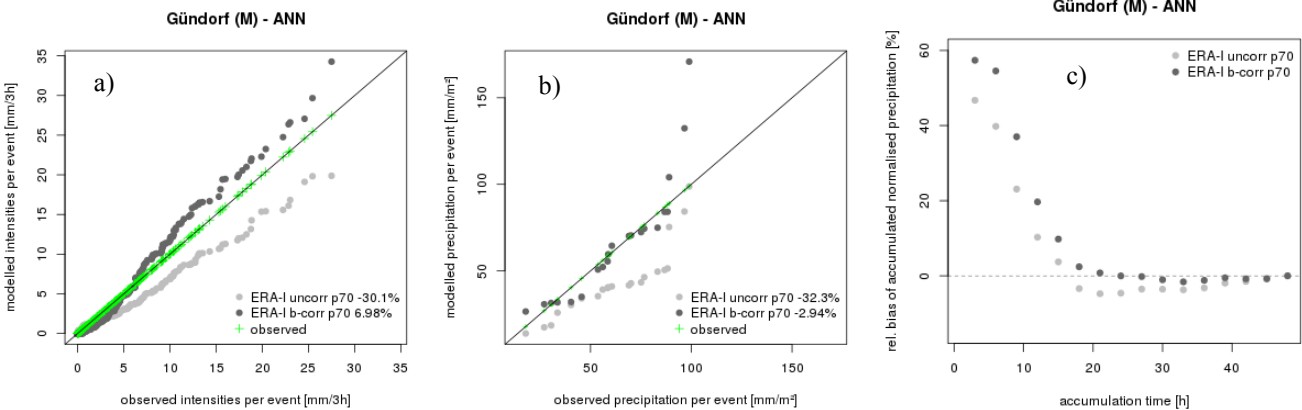

**Figure 12: Same as Figure 10, but for ERA-Interim in the catchment Gündorf.**

From a return period of 6-10 years the flood simulations are very sensitive to overestimations (e.g., gauges Voitsberg and
Gündorf in Figure 13) and underestimations (see Fig. 7) of the simulated rainfall, which is due to the non-linearity in the
rainfall-runoff process (e.g., Komma et al., 2007; Rogger et al., 2012). This threshold is consistent with usual concepts in
hydrology, such as the concept of the GRADEX method (e.g., Merz et al., 1999). At this size of floods the soils have been
saturated by a high amount of precipitation and 100% of the subsequent rainfall comes to runoff. This is vital to take into
account when it comes to correct high rainfall intensities within the bias correction procedure.

Seasonal occurrence is improved for all CCLM settings after bias correction (Figure 14). In particular, the shift from summer
to spring using the raw 0.11° and 0.44° data is removed. Again, the 3 km data yield results closest to the observed
distribution. Again, results using the bias corrected WRF data as input are incorporated into discussion in the synthesis step
in the following section.

As for the seasonality, the seasonal shift in the simulated soil moisture is removed after bias correction, but the
underestimation in summer and autumn cannot be entirely compensated (see supplementary material, Fig. S5 below). This
can be attributed to the fact that the modelled events are different in size, shape, and overall structure to those of
observations. The SDM methodology is performed independently for each grid cell, and as a result is not imposing the
structure of typical broad-scale observed weather events. Therefore, even though the distributions of bias corrected
precipitation align to observations at individual grid cells, the average precipitation amounts across multiple grid cells can
differ from observations. ERA-Interim results now lie exactly on the observation. However, for the MAF performance using
ERA-Interim data is not sufficient (compare Figure 13). This shows that using observed atmospheric conditions with large
grid size (~70 km) is able to reproduce mean monthly hydrological conditions, but fails in flood event representation on this
scale. Out of the CCLM data, performance using the 3 km data is still best, and underestimation using the 0.11° and 0.44°
data in summer is still evident in all catchments.

For an event based illustration of the effect of bias correction two events in 2009 at the Fluttendorf/Gnasbach gauge were chosen using the 3 km CCLM data as input (Figure 15). The first event in June is the largest in the series and the second event in August is the second largest in the series. Synoptic forcing is different between the two events: the first event is controlled by a persistent upper-air cut-off low that is located over the Balkan region and brings warm and moist air towards

the Eastern Alpine region from the East (Godina and Müller, 2009). This led to floods in the whole southern Styrian region, whereas the second flood is mainly driven by convective processes and concentrated on the eastern part. For the first event, the model with the uncorrected 3 km CCLM data simulates an event with the same order of magnitude, but slightly different timing, as the observation. After the bias correction flood peak is decreased due to a general reduction of precipitation in the bias correction in this period. A reduction of rainfall in this period is resulting from the bias correction as a consequence of

the overestimation of the MAFs by raw CCLM data (compare Fig. 7, upper right sub-plot). However, after bias correction, this is still the largest flood peak in the series (see Figure 13, upper right sub-plot). The second event is completely missed by the simulation run with the raw climate model data. No significant rainfall is simulated in the RCM and hence, bias correction is totally ineffective. It is clear, that at such missed events there is no possibility to correct raw RCM data by any statistical bias correction method. Bias correction is not able to compensate general uncertainties in representing convective

situations. Note that bias corrected intensities in upper panel are aggregated three hour sums.

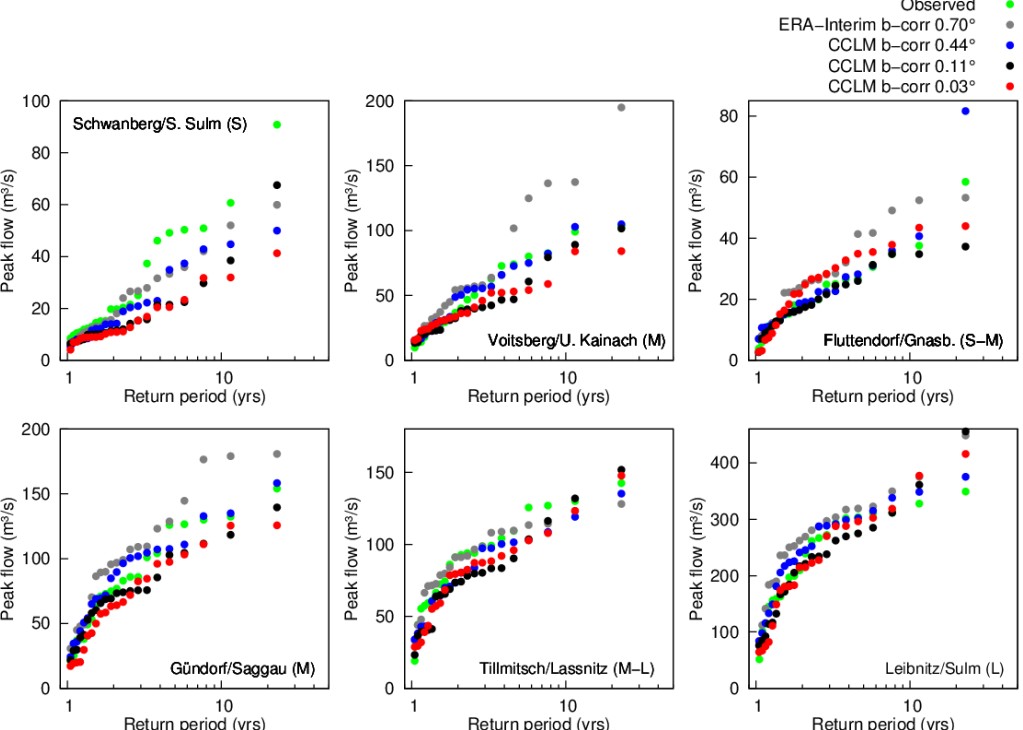

**Figure 13: Simulated maximum annual flood peaks using bias corrected CCLM data as input and observed maximum annual flood peaks vs. empirical return periods (Eq. (1), flood frequency plots) of the selected gauges in the period 1989-2010.**

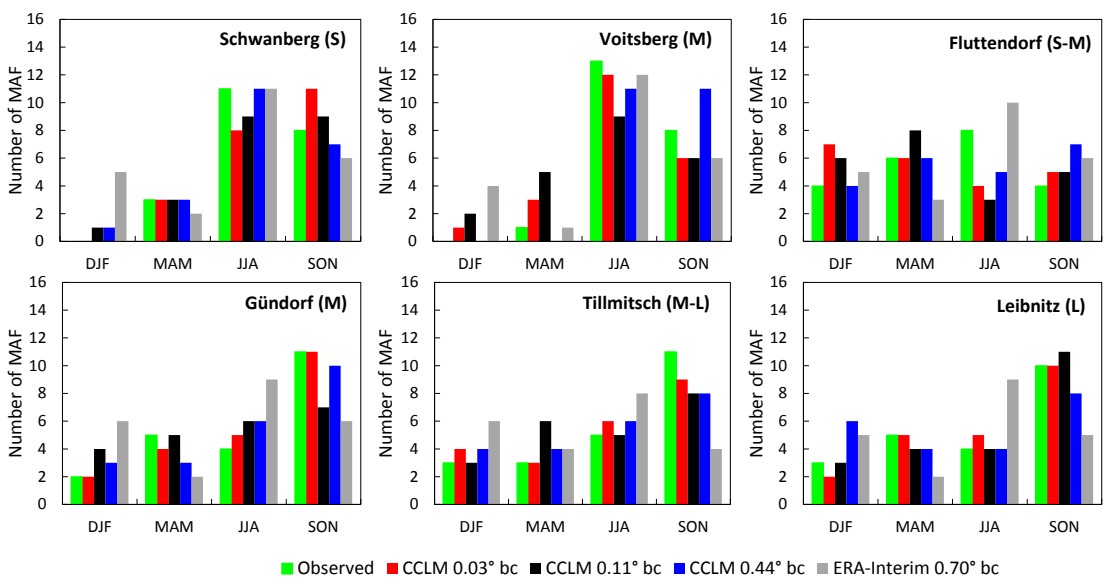

**Figure 14: Number of maximum annual floods in the four seasons (seasonality) from the simulation using bias corrected CCLM data as input compared to the observation at the selected gauges in the period 1989-2010.**

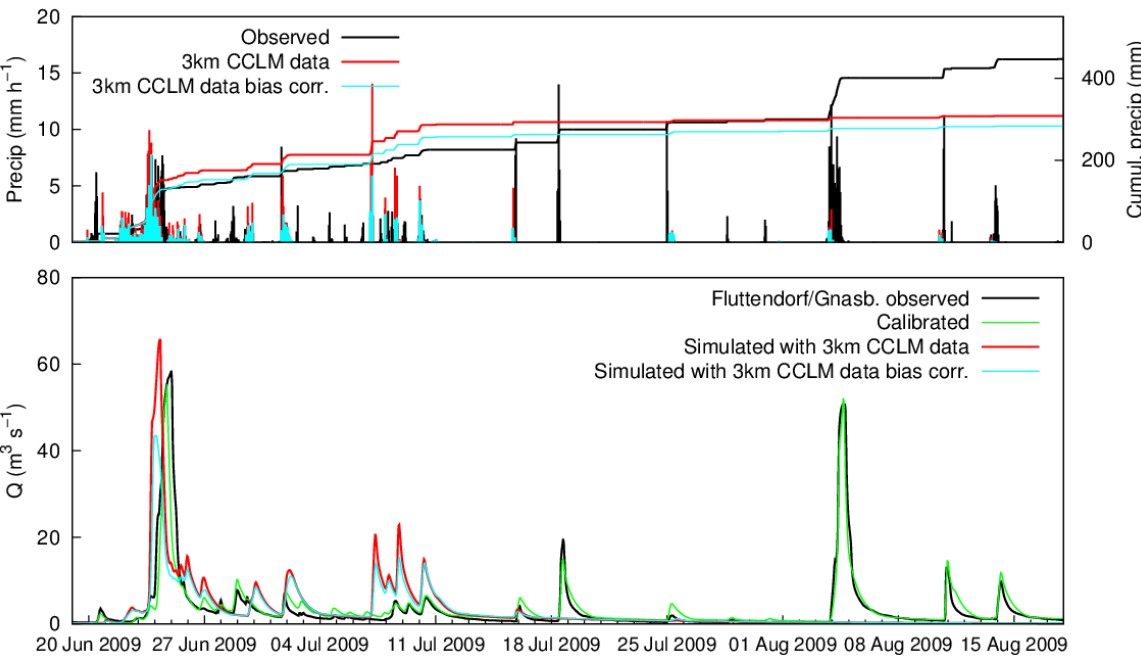

5 **Figure 15: Runoff simulated with the uncorrected (1h rainfall sums) and bias corrected (3h rainfall sums) 3km CCLM data for the period with the largest floods at the gauge Fluttendorf/Gnasbach. Above: catchment precipitation.**

### 6.3 Synthesis

The statistical measures of mean, standard deviation and skewness, for the 22 years sample of maximum annual floods resulting from the 14 different variants are illustrated in Figure 16. The mean (left plot column) and the standard deviation (middle plot column) are related to the catchment area in order to compare these measures between the gauges. Results using the ERA-Interim data are plotted in the centre, and the results using the different RCM settings with decreasing grid size are plotted towards the left (CCLM) and the right (WRF). The values with raw RCM data as input are plotted as black points; the values with bias corrected RCM data as an input are plotted as red points. The observed measures are indicated with a thin horizontal line for each gauge. The figure first clearly shows the decrease of mean specific runoff peaks and – in connection to this – the specific standard deviation with the catchment sizes (S to L from above) for all variants. This is mainly the consequence of a decrease of mean areal precipitation for large rainfall intensities and short durations (e.g., Hershfield, 1961; Lorenz and Skoda, 2000) but also of attenuation effects through flood routing. As discussed in the previous section, in most of the CCLM data driven simulations the statistical properties are improved reducing the grid size (black points) and further improved after bias correction (red points). For the larger catchments Tillmitsch and Leibnitz, the differences between the model variants are small, which, again, indicates the good performance of the coarser RCMs regarding general flood statistics (particularly CCLM). This improvement is not always the case for the WRF driven runs. Particularly large biases from the uncorrected run are either not compensated (e.g., WRF 0.44° for Schwanberg) or even over-compensated after bias correction (e.g., WRF 0.03° for Schwanberg and Voitsberg). The 3 km WRF produces in some periods unrealistic high rainfall intensities over several time steps, which leads to exceptional high flood peaks in the simulation. Examples are the very high values for the skewness (right plot column) at Gündorf and Voitsberg gauge. This high skewness could sometimes not be compensated after bias correction, e.g., Voitsberg gauge.

In order to summarize performance of the small multi-model ensemble regarding seasonality, the following Figure 17 shows the results applying Eq. (2) and Eq. (3-6) on the simulated MAFs using the different RCM data, raw (above) and after bias correction (below). The observation is plotted with a green filled square. As discussed in section 6.1, the results illustrate again the improvement of the seasonality using the 3 km CCLM data (full red squares) compared to the simulations with the coarser CCLM data for all gauges. For example, the highest concentration of timing, i.e. length of vector, of floods in a year in Voitsberg is represented well by the raw 3 km CCLM (upper middle sub-plot). However, this supreme result of CCLM 3km is the result of compensating errors: the complex interplay between single precipitation intensities and their temporal distribution during flood generating rainfall events is not correctly represented (section 6.2). Either the total precipitation amount is properly captured but the temporal distribution is failed or vice versa. This also holds for the other RCM simulations, including WRF 3km, and ERA-Interim. In addition, the bias correction method is not able to correct displacements in this complex interplay per construction and hence.

Using the coarser RCMs, both the timing and strength of seasonality of MAFs deviate significantly from the observations in all catchments. Moreover, the scatter between the different settings is large. However, to some extent all CCLM settings

represent the weak seasonality in the Eastern part (Fluttendorf catchment, upper right sub-plot). The convection-permitting WRF 3 km setting does not provide any improvements compared to the coarser resolutions. Timing of MAFs tends to be concentrated in May/June for all catchments, whereas flood events occur mainly from July to September. This indicates that more or less all WRF settings fail in representing the general mechanisms for flood generation in this area and at this scale.

5    Mostly, discrepancies can be compensated by the bias correction in the CCLM case, but not for the WRF case. In some catchments using the WRF 3 km settings the results are worse after bias correction. For example, at the Fluttendorf gauge (upper right sub-plot in Figure 17 below) the concentration of timing shifts from the beginning of May (with a low strength) to February (with a relatively high strength), a month when flood generation is also influenced by snow melt processes.

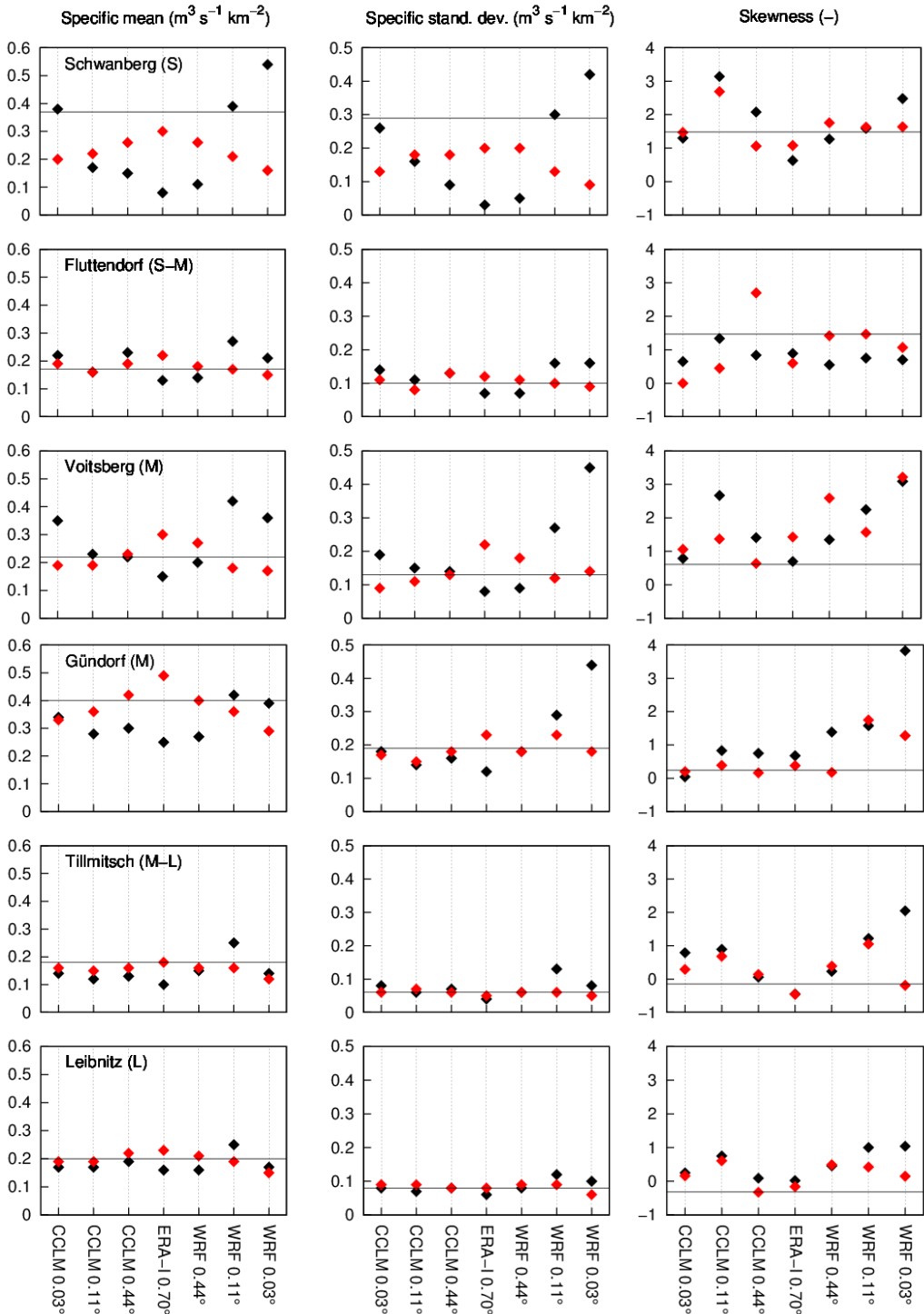

**Figure 16: Statistical measures of maximum annual flood peak distribution evolving the different model runs. Column 1: specific mean, column 2: specific standard deviation, column 3: skewness. Black: raw RCM data; red: bias corrected RCM data as input. Black horizontal line denotes the values from the observed flood peak series.**

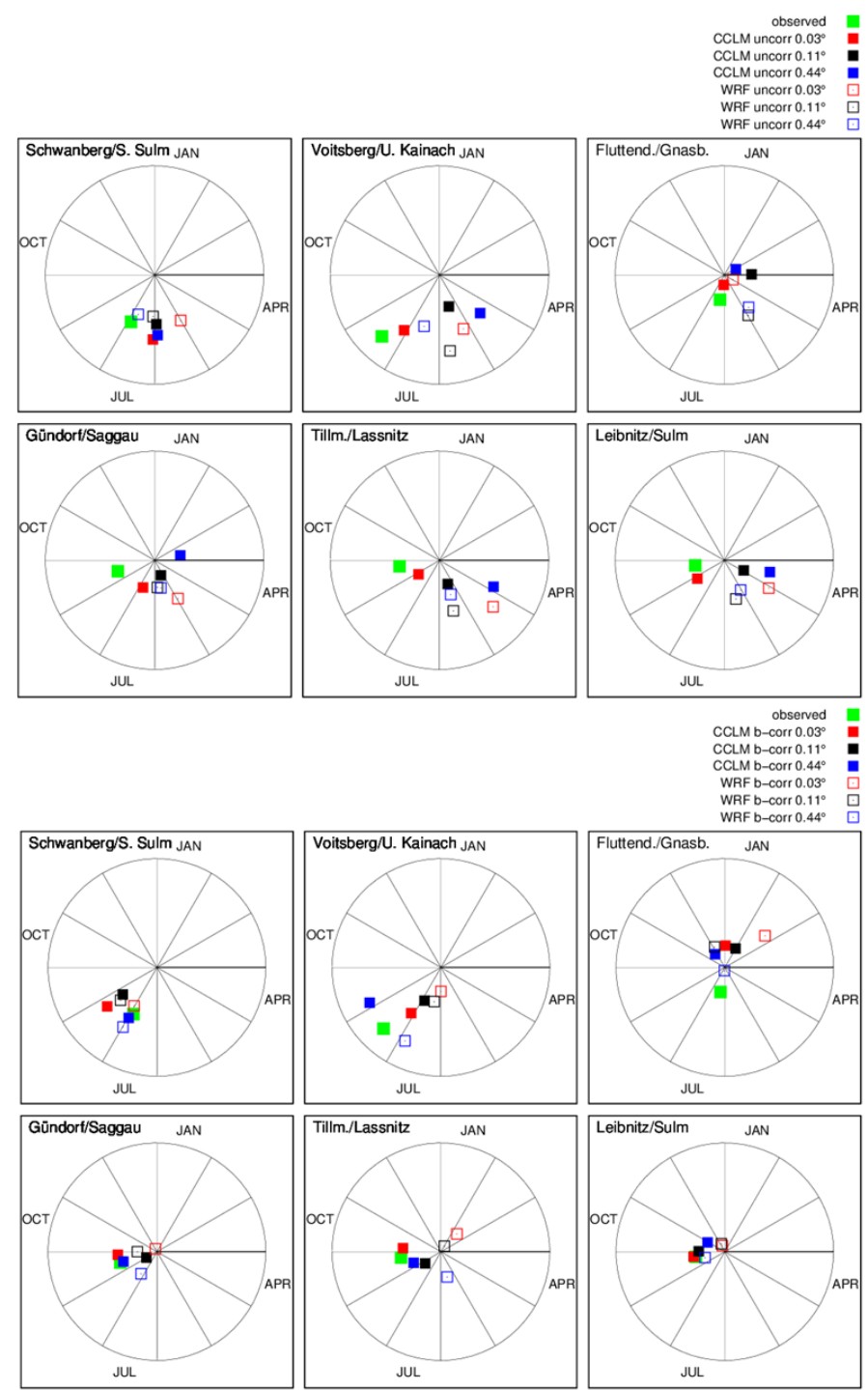

**Figure 17: Results of seasonality (circular statistics of the maximum annual floods) evolving the different model runs. Above: raw RCM data as input; below: bias corrected RCM data as input. The distance from the centre is the mean vector r and a measure for the seasonality strength, i.e. concentration of timing.**

**7 Conclusions**

This study implemented regional climate models sequentially coupled with a spatially distributed hydrological model to be used for enhanced flood modelling on small and medium spatial scales (up to app. 1000 km²) in the Eastern Alps. The work is carried out in a small multi-model (ensemble) framework using two different RCMs (CCLM and WRF) in different grid sizes, ~50 km, ~12.5 km including two runs at convection permitting scale (~3 km). Additionally, a novel bias correction method (i.e. a modified version of quantile mapping) is applied to minimize error propagation throughout the modelling chain. Together with the driving ERA-Interim data (grid size ~70 km) the ensemble contains 14 model variants.

Evaluations using observed data in a historical period (1989-2010) showed that in the investigated RCM ensemble, no clear added value of the usage of convection permitting RCMs for the purpose of flood modelling can be found, although CCLM 3 km outperforms in most flood statistics. This is based on the fact that flood events are the consequence of an interplay between the total precipitation amount per event and the temporal distribution of rainfall intensities on a sub-daily scale. The investigated RCM ensemble either lacks on one and/or the other. The seemingly good CCLM 3 km results in the small catchment lie on an overestimation of the intensities and underestimation of the total rainfall amount. This superposition is not systematic across the catchments. From a statistical perspective, all RCMs with all resolutions are able to produce precipitation rates that may cause floods in the study area.. In catchments with an area less than 100 km² a 1 hour time step due to the short response times is favourable but the influence is small. In the larger catchments, the 12.5 km and 50 km resolutions already yield satisfying results regarding flood statistics. However, with the coarser grid size the seasonality of floods, i.e. date of occurrence in a year, is not accurately represented. This indicates that some main flood generation mechanisms are not captured with the coarser models. CCLM 3km improves the seasonality of the maximum annual floods; however, in the light of the discrepancies mentioned above, the reason for this is not clear so far. An accurate representation of seasonality is important also in the light of recent findings by Blöschl et al. (2017) that shifts in the seasonality are the only consistent large-scale climate change signal regarding floods identified so far. The bias correction method Scaled Distribution Mapping (SDM) is able to systematically reduce biases on a seasonal basis. SDM improves results in magnitude and seasonality of maximum annual floods in all settings except for the small catchment (< 100 km²), which has to do with the intensity-rainfall amount interplay mentioned above.. The procedure corrects the rainfall amount but cannot correct the temporal dynamics. Also, due to the internal model variability, the RCM simulations partly decouple from their driving data and both synoptically forced and convective events may occur at different places or/and at different time as in the observations. Hence, in a usual climate modelling framework, i.e. long simulation periods and large RCM domain without nudging, an event-by-event analysis is not possible. Since the bias correction does not account for this effect and since it does not account for the number of sequential precipitation events (persistence), it might fail for single events and in weather type related approaches. Single events with very large biases – as seen using the WRF results – are over-compensated, i.e. an over-estimation is turned into an underestimation and vice versa. This affects the simulated flood peaks particularly for the higher return periods. The results further showed that the bias correction method is not able to compensate deviations in the

hydrological conditions, particularly in summer. This has implications on flood generation at summer storms, which are frequent in the study area, and highlights the need for further research regarding modifying rainfall events in this season within the bias correction.

With respect to climate change applications of convection permitting simulations for flood representation we can conclude that, despite the seemingly good results in the CCLM 3 km setting, attention has to be paid and the test of the results against historical data is of utmost importance. On the other hand, deep-convection parameterisations in coarser resolved standard RCMs have shown to be a source of "deep" uncertainty. For instance, Kendon et al. (2014) found significant increases in summertime precipitation in convection-permitting climate simulations in UK while the coarser resolved counterpart does not show any significant change. Ban et al. (2015) and Berthou et al. (2018) found similar results for short-term extreme precipitation events in the Alpine region and in the Mediterranean. In order to circumvent possibly misguided but far reaching climate change adaptation strategies, either convection permitting RCMs or proper statistical "convection emulators" (that are currently discussed in the climate modelling communities) should be used. Coarser models could still be used in larger catchments for rough estimations, but they should not be taken for granted regarding local/regional flood change. Also, there is a trade-off in the additional costs of a 3 km simulation and the postulated (small scale) process description as long as the physical representation of such small scale processes can be substituted by statistical ones. Regarding bias correction, the temporal dynamics of the rainfall have to be analysed; only if RCM errors are found to be systematic an application of a current error correction method can be recommended.

### Acknowledgments

The study was funded by the Austrian Climate Research Programme (ACRP 6th call, Proj.No. KR13AC6K11102 - CHC-FloodS) by the Austrian Climate and Energy Funds (KLIEN). Regional climate model output on the EURO-CORDEX domain was provided by Klaus Keuler (BTU Cottbus, Germany) and Klaus Görgen (Institute of Bio- and Geosciences, Agrosphere, Research Centre Jülich). The WRF simulations from K. Goergen used in this study were conducted at the Centre de Recherche Public -- Gabriel Lippmann, now: Luxembourg Institute of Science and Technology, under grant FNR C09/SR/16 (CLIMPACT) from the Luxembourg National Research Fund. The convection-permitting simulations were conducted in the course of the project NHCM-2 (nhcm-2.uni-graz.at), funded by the Austrian Science Fund (FWF, project no P24758-N29). We also thank Andreas F. Prein (National Center for Atmospheric Research, U.S.). He conducted the convection-permitting CCLM simulation and helped with fruitful comments. Computational resources were provided by the Jülich Supercomputing Centre (JSC) and by the Vienna Scientific Cluster (VSC). Additionally, we gratefully acknowledge the European Centre for Medium-Range Weather Forecasts (ECMWF) for providing ERA-Interim data. Hydro-meteorological data were provided by the Hydrographic Service of the province of Styria and the Central Institute for Meteorology and Geodynamics (ZAMG).

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

Tables

**Table 1 Stream gauges used for evaluation (Fig. 1).**

| Abbrev. in Fig. 1 | Gauge | River | Area (km²) | Data since |
|---|---|---|---|---|
| V | Voitsberg | Upper Kainach | 211 | 1966 |
| S | Schwanberg | Schwarze Sulm | 75 | 1951 |
| L | Leibnitz | Sulm | 1103 | 1951 |
| Gü | Gündorf | Saggau | 200 | 1982 |
| T | Tillmitsch | Lassnitz | 480 | 1961 |
| Fl | Fluttendorf | Gnasbach | 119 | 1968 |

5  **Table 2 RCMs and their settings.**

| Model | Grid size | Time step | nested in | conducted by |
|---|---|---|---|---|
| ERA-Interim | 0.7° (~70 km) | 6 h | - | ECMWF |
| CCLM 4.8 clm 17 | 0.44° (~50 km) | 1 h | ERA-Interim | WEGC (CORDEX) |
| CCLM 4.8 clm 17 | 0.11° (~12.5 km) | 3 h | ERA-Interim | BTU Cottbus (CORDEX) |
| CCLM 4.8 clm 17 | 0.03° (~3 km) | 1 h | CCLM 0.11° | WEGC (NHCM-2) |
| WRF 3.3.1 | 0.44° (~50 km) | 3 h | ERA-Interim | CRP-GL (CORDEX) |
| WRF 3.3.1 | 0.11° (~12.5 km) | 3 h | WRF 0.44° | CRP-GL (CORDEX) |
| WRF 3.3.1 | 0.03° (~3 km) | 1 h | WRF 0.11° | WEGC (NHCM-2) |

**Table 3 Model efficiency at the selected gauges in the calibration and historical (validation) period.**

| Gauge | Area (km²) | Calibration period 2000-2009 | | | Historical (validation) period 1989-1999 | | |
|---|---|---|---|---|---|---|---|
| | | BIAS (m³/s) | NSE (-) | RMSE (m³/s) | BIAS (m³/s) | NSE (-) | RMSE (m³/s) |
| Leibnitz/Sulm | 1103 | 0.85 | 0.88 | 5.86 | -0.28 | 0.83 | 8.15 |
| Tillmitsch/Lassnitz | 480 | 0.23 | 0.86 | 2.45 | -0.35 | 0.82 | 3.42 |
| Gündorf/Saggau | 200 | 0.28 | 0.84 | 1.93 | 0.39 | 0.56 | 3.28 |
| Schwanberg S. Sulm | 75 | 0.20 | 0.78 | 0.66 | 0.22 | 0.65 | 0.88 |
| Voitsberg/U. Kainach* | 211 | 0.15 | 0.83 | 1.26 | -0.13 | 0.85 | 1.33 |
| Fluttendorf/Gnasbach | 119 | 0.00 | 0.77 | 0.75 | -0.03 | 0.67 | 0.91 |

* Continuous runoff data since 1996 (only four years in historical period), but historical maximum annual flood peaks available (hydrographic year book)

