# Peer review of "Convection-permitting regional climate simulations for representing floods in small and medium sized catchments in the Eastern Alps"

_Natural Hazards and Earth System Sciences, 2018_

## Referee Comment (RC1) · Anonymous Referee #1 · 28 Mar 2018

**General comments**

Reszler et al. investigate the impact of spatial resolution in regional climate simulations on the capability of a hydrological model to represent the statistical characteristics of flood events in small catchments in the Eastern Alps. For this, ERA-Interim driven climate simulations with CCLM and WRF in different spatial resolutions (50km, 12.5km and 3km) are used as input data for the hydrological model KAMPUS. In a first step, KAMPUS is forced by raw model data, in a second step, by bias corrected simulation results. The statistical characteristics of the simulated floods are analyzed and compared to the measured runoff at stream gauges of six different catchments.

[Figure]

The study is very interesting, within the scope of NHESS and may merit publication. However, based on the presented results, I do not completely agree to the authors concluding statement that convection-permitting simulations are essential for a good flood representation. For uncorrected raw model data, this is obviously the case (figure 7, 8). But the bias corrected 50km simulation achieves, from my point of view, at least as good results as the uncorrected 3km run (figure 10, 11, 13, 14). This raises the question, whether a computationally expensive downscaling to 3km is necessary for a statistical consideration of floods or if bias correction of coarse data is sufficient? Based on the presented results, I would suggest that the improvements (if existing) of a downscaling to 3km do not justify its additional costs. I would recommend to put this question as central statement of the paper and thus, a major revision is needed.

A second interesting point of this study is that floods are only well represented in CCLM and not in WRF (figure 13, 14). This highlights the relevance of an adjusted RCM for each research area and should be mentioned and discussed more prominently.

specific comments

page 1, line 16: I would not say "ensemble" in this context, since the simulations are not really used as an ensemble. "Model chain" would be more appropriate.

page 1, line 21: I would use the term "coupling time step" to avoid confusion with the model time step.

page 6, line 9-11: The example is difficult to understand and should be rewritten. In general, the method of the bias correction should be described in more detail.

page 7, line 1: Why does KAMPUS use a temperature threshold to calculate the snow accumulation out of precipitation, instead of using directly the simulated snow from the RCM? CCLM 4.8_clm17 is known to have a cold bias in Winter, especially in 50km simulation (Kotlarski et al. 2014). In this way, this snow calculation method may lead to an overestimated snow amount in the hydrological model, resulting in a high snow melt

in spring which may cause the high flood peaks in spring in the CCLM 50km simulation. By correcting the cold bias in the CCLM results, this overestimated snow accumulation may be reduced, potentially explaining the improved seasonality in the bias corrected 50km simulation.

page 8, figure 3: What are the red areas?

page 10, figure 4: Why are you showing the average January precipitation amounts during night to highlight the added value of increasing model resolution? This is not the time frame in which I would expect the highest benefit from high resolution simulations (especially convection-permitting), but rather for summer (afternoon) precipitation.

page10, figure 5: The figure shows that the added value of an increased resolution is mainly caused by an improved diurnal cycle of precipitation. I would recommend to mention this more prominent, since this is very important for a realistic description of floods in smaller catchments.

page 11, line 7: Please add a reference for NSE.

page 12, figure 6: calibration and validation results should be drawn in different colors. In this way, it's difficult to assess the quality of the validation results.

page 24, conclusions: see above the general comments

References

Kotlarski, S., and Coauthors, (2014). Regional climate modeling on European scales: a joint standard evaluation of the EURO-CORDEX RCM ensemble. Geosci. Model Dev., 7, 1297–1333, doi:10.5194/gmd-7-1297-2014

---

## Referee Comment (RC2) · Anonymous Referee #2 · 10 Apr 2018

The manuscript investigates the added value of increased RCM resolution and bias correction to simulate localized heavy precipitation events over the Eastern Alps. Two different RCMs are tested (WRF and CCLM), both forced by ERA-Interim, and considering three nested domains at different resolutions (50km, 12.5 km and 3 km). The simulated precipitation fields are used as input data for a hydrological model over different sized catchments. The authors also investigate the added value of employing a bias correction technique to the hydrological model input data, using a Scaled Distribution Mapping method. The simulated results are compared against observational data in the 1989-2010 period, using different statistical measures. The study is generally interesting and well written. Nevertheless, I have some several concerns that I would

like the authors to address

Pg. 13, lines 8-11: In Figure 7 the improvement in using CCLM at 0.03° is very clear for the smallest catchment. But for the catchment with 119 km2, 0.11° obtains the best agreement, and even 0.70° seems closer to observations than 0.03° (except for the longest return period). In other catchments, coarser resolutions are also closer to observations. Stating that "simulations with coarser RCM data already yield reasonable results" is somewhat insufficient. Because increasing the resolution (to convective permitting) seems to degrade the flood frequency simulation in some cases (e.g. Voistsberg/U. Kanaish except for the highest return period, or Fluttendorf/Gnasb. compared to 0.11° or even 0.70°; Tillmitsch/Lassnitz and Leibniz./Sulm at intermediate return periods). This requires a more careful discussion. The presented results imply the need for a priori knowledge of the best resolution for flood frequency simulation in each catchment .

The improvement is even less clear when using WRF, which is given in supplementary material. On this matter, the choice of presenting WRF results as supplementary material is not clear to me, and I have some concerns about it. It is stated in the abstract that the manuscript is discussing two RCMs (and no further distinction is made between them is made). This is again repeated in the last paragraph of the introduction. The fact that the added value of convective-permitting resolution in WRF is lower, and often non-existent for both flood frequency and seasonality seems like a main result, given the aim of the proposed investigation.

Inspection of Figures 13 and 14 also raises major questions about the value of convective permitting resolution and bias-correction. The best resolution and whether bias-correction improves the results seems to vary significantly between different catchments.

Then in the conclusions it is stated that: "Flood frequency and seasonality is represented well in all catchments (. . .) However, the 3km grid size is essential for catchments smaller than 200 km2" This seems like an overstatement. For Fluttendorf (119km2), using CCLM uncorrected at 0.11° is better than 0.03° for simulation of flood frequency; and in the corrected case the essential nature of 3km is not clear at all. For WRF, 0.44° and 0.70° are better for uncorrected case considering flood frequency over Fluttendorf. For the corrected case, the essential nature of 3km for flood frequency is not evident. Seasonality in WRF is very often degraded by 0.03° resolution, for both corrected and uncorrected, including in the catchments with <200km2.

It is also stated in the conclusions that "in the larger catchments, the 12.5 km and 50 km resolution already yield satisfying results regards flood statistics". Concerning the flood frequency, the results are not "often not already satisfying", the problem here is that increasing to 0.03° degrades the results. Hence we need a priori knowledge of whether we should use convective permitting or not. For Seasonality, CCLM does seem to be improved by using 0.03°, but not WRF, which is also problematic. The abstract also reflects these unclear statements of added value, when compared to the results.

Minor Comments

- Pg. 1, Line 10: "an in increase in regional climate model". Instead of repeating "regional climate model", it could be replaced by RCM or just model.

- Pg. 1, Line 10: "Increase in regional climate model resolution and in particular, at the convection permitting scale, will lead to a better representation of the spatial and temporal characteristics of heavy precipitation at small and medium scales". This sentence is technically correct, but it's not very clear. It could be re-written. Increasing the resolution will lead to a better representation of small scales. But if we are using a coarser resolution the small scales are not represented (they are not explicitly simulated). Of course, this will depend on what is meant by "small and medium scales", which is not entirely clear. Perhaps quantify these. Notice that throughout the text "small and medium scales" is also used rather loosely. For example, in Pg. 3 line 13 it

is (30km2 to 1000 km2), but in pg. 8 line 17 it is (75kms to 200km2), while <1100 km2 is referred to as large.

- Pg. 24, line 10: "Moreover, catchments with an area less than 100 km2 require a 1-hour time step due to the short response times": this is based on one single case (Schwanberg)? Perhaps it should be stated that "Moreover, the catchement with an area. . .". It this generalizable?

- Figure 14 the green circle in the last panel (bottom right) is not visible.

---

## Referee Comment (RC3) · Anonymous Referee #3 · 17 Apr 2018

Title: Convection permitting regional climate simulations for representing floods in small and medium sized catchments in the Eastern Alps Authors: Reszler, C., Switanek, M.B., and Truhetz, H. Manuscript number: NHESS-2018-17

In this study the authors investigate the ability of regional climate models, run at convection permitting resolutions, to simulate localized flooding in small and medium catchments. Their methodology is to run a small ensemble of models at resolutions of 50km, 12.5km and 3km the output from which are fed into a distributed hydrological model. The hydrological model is shown to be quite accurate when calibrated and tested. They find that value is mostly added for the higher resolutions over the smallest catchments (<200km^2) and sub-daily time scales (e.g. hourly). Despite these improvements in performance bias adjustment is still required. However, due to the temporal smoothing of the bias adjustment the higher resolution does not always yield improved results. Also, the results are highly dependent on the driving regional climate model. The CCLM results are far superior to those obtained with the WRF modeling system.

This study represents an important step in the application of convection permitting models. Much of the promise of these platforms rests on their ability to drive downstream impacts models and it is nice to see them applied in this manner. The authors do a good job demonstrating that there is significant added value in modeled precipitation at sub-daily time scales (e.g. figures 4 and 5) and that the flood representation (both maximum and seasonality) is generally, though not always, improved, at least for the medium to small catchments. Unfortunately applying the bias adjustment degrades the performance over the smaller catchments for the convection permitting simulations. While the bias adjustment improves the outputs from the 50km and 12.5km simulations it is difficult to see any improvement in the 3km simulations. I understand that there is some degradation due to the temporal smoothing but then one must ask: why bother?

I think this makes an important contribution but it raises more questions than it answers and as such I think the authors need some more nuanced discussion and also to tone down the conclusions a bit. The conclusion that ∼3km is "essential" for catchments smaller than 200km is overstating the fact as only two catchments are below 200km and neither shows marked improvement in either maximum flood peaks (figure 10) or seasonality (figure 11) once the bias adjustment is applied. Also the sample of two catchments is too small to make any generalizable conclusions. Maybe the authors could add a few more small catchments to the study to help bolster their case. Also there is the fact is that these results appear to be highly model dependent. The authors offer some explanation by way of the fact that CCLM is well tuned and widely used over this region while WRF is not. However, recent studies show comparable performance

by WRF over the Greater Alpine Region and Europe more generally (Awan et al., 2015; Knist et al., 2018; Kunstmann et al. 2018). Rather than a hand waving generalization about model family rather an more detailed description of which processes the simulations reproduce correctly and why might be more informative.

What would help greatly would be a more in-depth look at the bias correction method, discussion of the effects of the different nesting strategies, the addition of more small catchments and more nuanced discussion and conclusion sections. The bias adjustment technique is described as "novel" and as such readers may not be as familiar with it as they are with more common quantile mapping approaches. The use of a non-stationary approaches is well justified however the reader needs some more information especially in light the rather modest improvement the bias adjustments affords. Also the conclusion section should be rewritten with a more nuanced interpretation of the results. Is convection permitting modeling really needed if, after bias adjustment, results are no better or only modestly improved compared to coarser resolution simulations? Should multi-model, multi-realization, ensembles be employed or rather one highly tuned simulation? For present climate this might be sufficient but such tuning has well demonstrated shortcomings at climate time scales. Clearly there are substantial challenges remaining before these types of simulations can reliably be used for impacts models. At present the authors fail to acknowledge this and I think somewhat overstate their results. Also, the authors claim that recommendations can be made but then fail to deliver on this promise. What general recommendations, if any, can be made based on this study? Does convection permitting modeling only provide added value over particular areas, for particular cases, particular time scales and particular cases/phenomena? The results here certainly seem to point towards such a limited use or at the very least a need to balance expectations with current capabilities. I recommend a major revision as there is considerable additional discussion/clarification needed and potentially additional analysis is required. Specific comments follow below.

Specific Comments P3L7-8: The community is well beyond "first attempts". WRF-

Hydro (a fully coupled distributed hydrological model within WRF) is far enough in its development to be the core model for the United States' National Water Model. https://ral.ucar.edu/projects/wrf_hydro/overview. Other such systems in operation are TerrSysMP which features a 3-D groundwater model coupled to COSMO-CLM (e.g. Keune et al., 2017). Note that I do not make any on their reliability over climate time scales (i.e. simulations around a decade or more).

P3L13: Please be clear that when you write "coupled" you mean limited one-way coupling (I actually wouldn't call this coupling at all) wherein there is no feedback between the hydrological model and the atmospheric model and the atmospheric model only passes temperature and precipitation.

P4 L1-7: What are the potential ranges of observational uncertainty? In addition to sensor errors and under catch there is also uncertainty resulting from interpolating to a grid from point based station data. How are these taken into account?

P4L8: The version of WRF used here is quite old (almost 8 years!) and many of the issues related to this version have been corrected. In fact WRF is now 6 full versions more advanced as of this writing. How might this have affected to the results?

P6-Error Correction: More details on the bias adjustment are needed given that it is being pitched as a "novel" approach. How does it perform relative to other approaches? What are its limitations and/or tradeoffs? What are the implications of univariate approach to bias adjustments when the two variables corrected, temperature and precipitation, are related to each other? Also more explanation of the issues/limitations behind current approaches is needed. Maraun et al (2017) have an excellent overview of the current state of bias adjustment shortcomings and placing the SDM approach among these would be helpful to readers.

P8L29: "relatively".

P12L1: Specify which figure/panel you are referring to.

P13L10: I don't believe it has been demonstrated that CPM is "absolutely necessary". I would suggest either making a stronger argument with stronger supporting evidence or modify this claim.

P17L23: The authors write that "performance using 3km data is still best" but it is hard to discern this from the figures. Figure 11 show 24 seasons in total and of those the 3km is closest to observations in only 11 of these. In the other seasons either the 0.11 or 0.44 degrees simulation is closer to observations or the performance across resolutions is equal.

P16L10: Performance after bias adjustment is degraded over the smallest catchment. Yet this is precisely the type of application (i.e. small catchments) where the authors argue we see the greatest added value of convection permitting modeling. Here it appears that the two techniques, high-resolution modeling and bias adjustment, are not working in concert but in opposition.

P20L30-31: This is in direct contradiction to earlier, and later statements, that CP scales are "absolutely necessary". I would say rather that it is clear that there is still quite some work to do before these models can be reliably used for these sorts of applications.

P24L10: See previous comment. I do not think the authors have presented evidence sufficient to make this statement.

P25L17-18: Clearly CCLM has higher performance than WRF in this study. However, the authors never discussed the nesting strategy (see table 2), which is different for each model system. Specifically, WRF goes through an additional intermediate nest, a step that will certainly have an impact. How then are then are the WRF and CCLM simulations directly comparable?

P25L4: What "recommendations", specifically, can be made?

P25L5: The modeling systems used here are not "coupled" they are used in a model

chain.

Figures and tables

Figure 1. It is almost impossible to make out the catchment boundaries and initials in this busy figure. I suggest moving the catchment labeling to the larger figure 3.

Figure 3. Place catchment labels here in bold. Also bold lines around the catchments themselves so that the six catchments under investigation are clearly delineated.

Figure 4. What region is shown here?

Figure 5. Remove the empty panel in the lower right corner.

Figure 7. It is very hard to distinguish between the blue and black circles. Also including red and green is not colorblind friendly. I suggest a different color scale that has greater separation. This comment applies to Figures 7-12 and 14.

References

Awan, N. K., Gobiet, A., & Suklitsch, M. (2015). The role of regional climate model setup in simulating two extreme precipitation events in the European Alpine region. Climate Dynamics, 44(1–2), 299–314. https://doi.org/10.1007/s00382-014-2323-1

Keune, J., Gasper, F., Goergen, K., Hense, A., Shrestha, P., Sulis, M., & Kollet, S. (2016). Studying the influence of groundwater representations on land surface‐atmosphere feedbacks during the European heat wave in 2003. Journal of Geophysical Research: Atmospheres, 121(22). https://doi.org/10.1002/2016JD025426

Knist, S., Goergen, K., & Simmer, C. (2018). Evaluation and projected changes of precipitation statistics in convection-permitting WRF climate simulations over Central Europe. Climate Dynamics, 1-17. https://doi.org/10.1007/s0038

Kunstmann et al. (2018) Very High Resolution Regional Climate Simulations for Germany and the Alpine Space: Optimized Model Setup, Perfor-
mance in High Mountain Areas and Expected Future Climate. Geophysical Research Abstracts Vol. 20, EGU2018-8505, 2018 EGU General Assembly 2018 https://meetingorganizer.copernicus.org/EGU2018/EGU2018-8505.pdf

Maraun, D., Shepherd, T. G., Widmann, M., Zappa, G., Walton, D., Gutiérrez, J. M., . . . Mearns, L. O. (2017). Towards process-informed bias correction of climate change simulations. Nature Climate Change, 7(11), 664–773. https://doi.org/10.1038/nclimate3418

---

## Author Comment (AC1) · 7 Jun 2018

**Author's response (AR)**

**Response to reviewer 1**

*Reviewer #1 general comments:*

*RC: But the bias corrected 50km simulation achieves, from my point of view, at least as good results as the uncorrected 3km run (figure 10, 11, 13, 14). This raises the question, whether a computationally expensive downscaling to 3km is necessary for a statistical consideration of floods or if bias correction of coarse data is sufficient? Based on the presented results, I would suggest that the improvements (if existing) of a downscaling to 3km do not justify its additional costs. I would recommend to put this question as central statement of the paper and thus, a major revision is needed.*

AR: We agree that results of the model chain using the coarser RCM together with error correction procedure look good. However, as the uncorrected RCM results show, seasonality is wrong in the coarser models, which indicates a lack in capturing the main atmospheric mechanisms for flood generation. From the point of applying a climate impact model chain, bias correction should either (1) only correct (small) biases, i.e. systematic errors, and not compensate errors in process description in order to prevent from the 'model is right for the wrong reasons' case (Klemes 1986) or (2) make use of process-informed approaches (e.g. Maraun et al., 2017) that are currently discussed in the climate modelling communities but are far from being established. Moreover, as it was recently shown by Blöschl et al. (2017), shifts in the seasonality are the only consistent large-scale climate change signal regarding floods identified so far. Our bias correction is largely compensating the improper representation of seasonality in the coarser models, however this is done in a statistical manner and we cannot exclude that we are still doing the 'right for the wrong reasons'. On the other hand, convection-permitting models gain from high resolutions and numerically resolved deep convective processes. This represents a fundamental change in the modelling technique which can have a substantial impact on projected climate change effects. For instance, Kendon et al. (2014) found significant increases in summertime precipitation in convection-permitting climate simulations in UK while the coarser resolved counterpart does not show any significant change. Ban et al. (2015) and Berthou et al. (2018) found similar results for short-term extreme precipitation events in the Alpine region and in the Mediterranean. However, since such simulations are relatively new, their benefits and shortcomings in climate applications are largely unknown, especially their potential in flood-modelling has not been explored. Nevertheless in the conclusion, we will reduce the strength of the requirement to use a 3 km model, since it also hinges on the partly wrong coarser models as its driving data and biases propagate along the downscaling chain (e.g. Addor et al., 2016). But the first results are promising. We will also include a statement in the conclusions that so far the coarser models could be used for climate impact studies in larger catchments for rough estimations, but they should not be taken for granted regarding local/regional flood change. We agree that so far, there is a trade-off in the additional costs of a 3 km simulation and the postulated (small scale) process description as long as the physical representation of such small scale processes can be substituted by statistical ones. We will include this trade-off question into the introduction, discussion of the simulation runs and conclusions. We will therefore extend the synthesis chapter by a comprehensive discussion. It was not the focus of the study; that would

also include cost-benefit analyses, also monetarily. We believe/hope that computational infrastructure and efficiency will further improve to reduce these costs.

Addor, N., Rohrer, M., Furrer, R. and Seibert, J.: Propagation of biases in climate models from the synoptic to the regional scale: Implications for bias adjustment, J. Geophys. Res.-Atmospheres, 121(5), 2075–2089, doi:10.1002/2015JD024040, 2016.

Ban, N., Schmidli, J. and Schaer, C.: Heavy precipitation in a changing climate: Does short-term summer precipitation increase faster?, Geophys. Res. Lett., 42(4), 1165–1172, doi:10.1002/2014GL062588, 2015.

Berthou, S., Kendon, E. J., Chan, S. C., Ban, N., Leutwyler, D., Schär, C. and Fosser, G.: Pan-European climate at convection-permitting scale: a model intercomparison study, Clim. Dyn., doi:10.1007/s00382-018-4114-6, 2018.

Blöschl et al.: Changing climate shifts timing of European floods. Science, 357 (2017), 6351; 588 – 590, 2017.

Kendon, E. J., Roberts, N. M., Fowler, H. J., Roberts, M. J., Chan, S. C. and Senior, C. A.: Heavier summer downpours with climate change revealed by weather forecast resolution model, Nat. Clim. Change, 4(7), 570–576, doi:10.1038/NCLIMATE2258, 2014.

Klemeš, V.: Operational testing of hydrological simulation models. Hydrological Sciences Journal, 31(1), 13-24, 1986.

Maraun, D., Shepherd, T. G., Widmann, M., Zappa, G., Walton, D., Gutiérrez, J. M., Hagemann, S., Richter, I., Soares, P. M. M., Hall, A. and Mearns, L. O.: Towards process-informed bias correction of climate change simulations, Nat. Clim. Change, 7(11), 664–773, doi:10.1038/nclimate3418, 2017.

*RC: […] floods are only well represented in CCLM and not in WRF (figure 13, 14). This highlights the relevance of an adjusted RCM for each research area and should be mentioned and discussed more prominently.*

AR: We agree, CCLM and WRF show different behaviour depending on resolution and research area which asks for a bias correction. We will include a more rigorous analysis of the effects and the applicability of our bias correction technique in flood modelling attempts. Especially, since the bias correction method does not affect the frequency of precipitation it is rather unclear at the current stage how the statistical correction of precipitation intensities affect flood events that rely on a correct representation of the precipitation sequence and their occurrence in a climatological sense.

*Reviewer #1 specific comments:*

*RC: page 1, line 16: I would not say "ensemble" in this context, since the simulations are not really used as an ensemble. "Model chain" would be more appropriate.*

AR: We agree and will change this.

*RC: page 1, line 21: I would use the term "coupling time step" to avoid confusion with the model time step.*

AR: We agree and will change this.

*RC: page 6, line 9-11: The example is difficult to understand and should be rewritten. In general, the method of the bias correction should be described in more detail.*

AR: Since the new error correction method has been published recently, we didn't include a comprehensive description. But we will do this for a better understanding.

*RC: page 7, line 1: Why does KAMPUS use a temperature threshold to calculate the snow accumulation out of precipitation, instead of using directly the simulated snow from the RCM? CCLM 4.8_clm17*

AR: For consistency with the calibration we will stick to this simple model. This is widely used in flood forecasting to avoid additional uncertainties introduced by the use of highly variable climate variables from weather/climate models and therefore increase robustness of the models. Same with evapotranspiration.

*RC: [..] By correcting the cold bias in the CCLM results, this overestimated snow accumulation may be reduced, potentially explaining the improved seasonality in the bias corrected 50km simulation.*

AR: Yes, this is captured in the bias correction by correcting air temperature. However, the shift in runoff seasonality (overestimation of runoff in spring and underestimation in summer) is the consequence of the (same) shift in precipitation. We will add the precipitation distribution from the different RCMs over a year (monthly basis) in the suppl. material.

*RC: page 8, figure 3: What are the red areas?*

AR: The figure will be redrawn. Layout is misunderstanding. The blue colour is used to denote nested catchments within the larger ones (in red).

*RC: page 10, figure 4: Why are you showing the average January precipitation amounts during night to highlight the added value of increasing model resolution? This is not the time frame in which I would expect the highest benefit from high resolution simulations (especially convection-permitting), but rather for summer (afternoon) precipitation.*

AR: Figure 4 is only an example. We decided to remove it, since it does not contain any additional information needed for the explanation in the text.

*RC: page10, figure 5: The figure shows that the added value of an increased resolution is mainly caused by an improved diurnal cycle of precipitation. I would recommend to mention this more prominent, since this is very important for a realistic description of floods in smaller catchments.*

AR: We agree. This will be mentioned together with a better explanation of the SDM method.

*RC: page 11, line 7: Please add a reference for NSE.*

AR: Will be done.

*RC: page 12, figure 6: calibration and validation results should be drawn in different colors. In this way, it's difficult to assess the quality of the validation results.*

AR: Will be done.

*RC: page 24, conclusions: see above the general comments*

AR: A comprehensive discussion section will be included (see above).

**Response to reviewer 2**

*Reviewer #2 general comments:*

*RC: Pg. 13, lines 8-11: In Figure 7 the improvement in using CCLM at 0.03° is very clear for the smallest catchment. But for the catchment with 119 km2, 0.11° obtains the best agreement, and even 0.70° seems closer to observations than 0.03° (except for the longest return period). In other catchments, coarser resolutions are also closer to observations. Stating that "simulations with coarser RCM data already yield reasonable results" is somewhat insufficient. Because increasing the resolution (to convective permitting) seems to degrade the flood frequency simulation in some cases (e.g. Voistsberg/U. Kanaish except for the highest return period, or Fluttendorf/Gnasb. compared to 0.11° or even 0.70°; Tillmitsch/Lassnitz and Leibniz./Sulm at intermediate return periods). This requires a more careful discussion. The presented results imply the need for a priori knowledge of the best resolution for flood frequency simulation in each catchment.*

> AR: We will add a comprehensive discussion with reduced strength regarding the need of CPS and the benefit of the bias correction. Improvement of CPS is not evident in every case regarding flood frequency, but it is evident for seasonality (see response to general comments of reviewer #1).

*RC: The improvement is even less clear when using WRF, which is given in supplementary material. On this matter, the choice of presenting WRF results as supplementary material is not clear to me, and I have some concerns about it. It is stated in the abstract that the manuscript is discussing two RCMs (and no further distinction is made between them is made). This is again repeated in the last paragraph of the introduction. The fact that the added value of convective-permitting resolution in WRF is lower, and often non-existent for both flood frequency and seasonality seems like a main result, given the aim of the proposed investigation.*

> AR: We focused on the CCLM results for explaining the evaluation procedure in order to avoid a too long paper. Also, during the study starting in 2013 we had problems with the WRF simulation. For a long time it was not clear if we receive results that can be interpreted like the CCLM results (crash, bug, etc., see specific response to reviewer #3, p.10). We will add a deeper discussion with possible reasons why the WRF 3km more or less fails in representing floods (processes, nesting, etc.).

*RC: Inspection of Figures 13 and 14 also raises major questions about the value of convective permitting resolution and bias-correction. The best resolution and whether bias correction improves the results seems to vary significantly between different catchments.*

*Then in the conclusions it is stated that: "Flood frequency and seasonality is represented well in all catchments. However, the 3km grid size is essential for catchments smaller than 200 km2. This seems like an overstatement. For Fluttendorf (119km2), using CCLM uncorrected at 0.11° is better than 0.03° for simulation of flood frequency; and in the corrected case the essential nature of 3km is not clear at all. For WRF, 0.44° and 0.70° are better for uncorrected case considering flood frequency over Fluttendorf. For the corrected case, the essential nature of 3km for flood frequency is not evident. Seasonality in WRF is very often degraded by 0.03° resolution, for both corrected and uncorrected, including in the catchments with <200km2.*

*It is also stated in the conclusions that "in the larger catchments, the 12.5 km and 50 km resolution already yield satisfying results regards flood statistics". Concerning the flood frequency, the results are not "often not already satisfying", the problem here is that increasing to 0.03° degrades the results. Hence we need a priori knowledge of whether we should use convective permitting or not. For Seasonality, CCLM does seem to be improved by using 0.03°, but not WRF, which is also problematic. The abstract also reflects these unclear statements of added value, when compared to the results.*

> AR: This is true, there is no systematic behaviour. Any differences are further amplified by the non-linearity in the flood generation process (particularly above return periods of 5-8 years, small differences in precipitation can induce large differences in flood peaks). We will include this into the discussion. On the other hand, this shows the importance of a test of the models against historical data before applying in impact analyses.

*Reviewer #2 minor comments:*

*RC: Pg. 1, Line 10: "an in increase in regional climate model". Instead of repeating "regional climate model", it could be replaced by RCM or just model.*

> AR: OK.

*RC: Pg. 1, Line 10: "Increase in regional climate model resolution and in particular, at the convection permitting scale, will lead to a better representation of the spatial and temporal characteristics of heavy precipitation at small and medium scales". This sentence is technically correct, but it's not very clear. It could be re-written. Increasing the resolution will lead to a better representation of small scales. But if we are using a coarser resolution the small scales are not represented (they are not explicitly simulated). Of course, this will depend on what is meant by "small and medium scales", which is not entirely clear. Perhaps quantify these. Notice that throughout the text "small and medium scales" is also used rather loosely. For example, in Pg. 3 line 13 it is (30km2 to 1000 km2), but in pg. 8 line 17 it is (75kms to 200km2), while <1100 km2 is referred to as large.*

> AR: The first sentence will be re-written. We take out the term "small and medium scales" here, in order to avoid the mismatch of terms from modelling techniques ("processes on resolved/unresolved scales") and the size of the investigated catchments ("small and medium"). The new version of the sentence now is: "Regional climate model (RCM) evaluations and inter-comparisons have shown that an Increase in regional climate model resolution and in particular, at the convection permitting scale, will lead to a better representation of the spatial and temporal characteristics of heavy precipitation." Further on, the terms "small and medium scales" will be clarified and used consistently through the text.

*RC: Pg. 24, line 10: "Moreover, catchments with an area less than 100 km2 require a 1-hour time step due to the short response times": this is based on one single case (Schwanberg)? Perhaps it should be stated that "Moreover, the catchment with an area …". It this generalizable?*

> AR: We tested the influence of the time step by using the 3hr sums of the CCLM 3km and comparing to the 1 hr results. There is a decrease of flood peaks, but the main decrease in performance in the small catchment Schwanberg is due to the error correction. We will discuss

the reasons more in detail on the basis of a better explanation of the SDM method (see response to reviewer #1).

*RC: Figure 14 the green circle in the last panel (bottom right) is not visible.*

AR: It is behind the red-filled square. The symbol for the observations will be made larger.

**Response to reviewer 3**

*Reviewer #3 general comments:*

*RC: [..] but it raises more questions than it answers and as such I think the authors need some more nuanced discussion and also to tone down the conclusions a bit. [..]*

AR: We will include a comprehensive discussion about the different results and reduce the strength of the conclusion (see response to reviewer #1). We agree, that there are still some open questions which we will address (some questions "OQ" are marked later in the text).

*RC: [..] Also the sample of two catchments is too small to make any generalizable conclusions. Maybe the authors could add a few more small catchments to the study to help bolster their case. [..]*

AR: Of course, the extent of the test area in south eastern Austria cannot claim any generalized conclusions for say, the European scale. It is an area, where a spatially distributed model could be calibrated on a very small scale sub-catchment basis. The evaluation catchments were carefully selected in order to be representative in the area. As stated in the text, despite the small overall test area extent, the variety of climatic, topographic, geologic and pedologic properties is high. The catchments were selected to represent these different properties. There are several gauges in neighbouring or nested catchments available and the results there are consistent with the selected catchment for each region. **OQ:** Perhaps this small scale variability of catchment properties (which lead to different model parameters – response times, non-linearity!) is one reason for the unsystematic results. Also, the quality of some temporal characteristics of the catchment-accumulated precipitation (frequency, duration, intensity) that is simulated by CCLM and WRF in their various resolutions has not been investigated yet. We will address these issues (see also response to reviewer #1).

*RC: Also there is the fact is that these results appear to be highly model dependent. The authors offer some explanation by way of the fact that CCLM is well tuned and widely used over this region while WRF is not. However, recent studies show comparable performance by WRF over the Greater Alpine Region and Europe more generally (Awan et al., 2015; Knist et al., 2018; Kunstmann et al. 2018). Rather than a hand waving generalization about model family rather an more detailed description of which processes the simulations reproduce correctly and why might be more informative.*

AR: Yes, we agree. The generalization is too short-sighted. It has been demonstrated multiple times, that CCLM and WRF show similar performance indices for precipitation on coarser resolutions (12.5 km, 50 km), e.g. Kotlarski et al. (2014), Smiatek et al. (2016). However, when it comes to convection-permitting simulations, a judgement of the model performance becomes

difficult, because such systematic model-inter-comparison studies do not exist and one has to rely on published evaluation studies, that vary in the model domain and/or in the length of the simulation period. For instance, Knist et al. (2018) conducted and compared a pan-European WRF simulation with 3 km horizontal grid spacing with data records from ground based stations, however, the majority of the stations covers Germany and only a few of them are located in the Alpine region. On the other hand, Ban et al. (2014) evaluated a CCLM simulation with 2.2 km grid spacing, but only with station data from Switzerland, which raises questions about the comparability with the results of Knist et al. (2018). Nonetheless, both studies have in common, that the convection permitting simulations capture the frequencies of heavy and extreme hourly precipitation better than their coarser resolved counterparts and, that extreme events are more overestimated in mountainous regions. We agree, that our CCLM and WRF simulations are ideal to fill the gap of missing comparable convection permitting simulations; however, the focus of this paper lies on driving a flood model with RCMs and to investigate the effects of the RCMs' resolutions and a bias correction on the representation of floods. For the time being, we will put more effort on understanding the effects of the bias correction (see response to reviewer #1) and provide understating of the sources of RCM biases only as far as it is necessary. A thorough CCLM/WRF inter-comparison study is out of the scope of this paper. Such systematic model-inter-comparison studies are subject to the Flag Ship Pilot Study (FPS) "Convective phenomena at high resolution over Europe and the Mediterranean" of the Coordinated Downscaling Experiment (CORDEX) of the World Climate Research Programme (WCRP) that has started in 2016 and to which CCLM and WRF simulations of the Wegener Center contribute to.

Ban, N., Schmidli, J. and Schaer, C.: Evaluation of the convection-resolving regional climate modeling approach in decade-long simulations, J. Geophys. Res.-Atmospheres, 119(13), doi:10.1002/2014JD021478, 2014.

Knist, S., Goergen, K. and Simmer, C.: Evaluation and projected changes of precipitation statistics in convection-permitting WRF climate simulations over Central Europe, Clim. Dyn., doi:10.1007/s00382-018-4147-x, 2018.

Kotlarski, S., Keuler, K., Christensen, O. B., Colette, A., Déqué, M., Gobiet, A., Goergen, K., Jacob, D., Lüthi, D., van Meijgaard, E., Nikulin, G., Schär, C., Teichmann, C., Vautard, R., Warrach-Sagi, K. and Wulfmeyer, V.: Regional climate modeling on European scales: a joint standard evaluation of the EURO-CORDEX RCM ensemble, Geosci. Model Dev., 7(4), 1297–1333, doi:10.5194/gmd-7-1297-2014, 2014.

Smiatek, G., Kunstmann, H. and Senatore, A.: EURO-CORDEX regional climate model analysis for the Greater Alpine Region: Performance and expected future change: CLIMATE CHANGE IN THE GAR AREA, J. Geophys. Res. Atmospheres, 121(13), 7710–7728, doi:10.1002/2015JD024727, 2016.

*RC: What would help greatly would be a more in-depth look at the bias correction method, discussion of the effects of the different nesting strategies, the addition of more small catchments and more nuanced discussion and conclusion sections. […]*

AR: A more comprehensive description of the bias-correction method and how it alters precipitation events that cause floods in our catchments will be given (see response to reviewer #1). Concerning nesting: CCLM 0.44° and CCLM 0.11° are directly driven by ERA-Interim (single nesting), while WRF 0.11° is nested into WRF 0.44° (double nesting). Note, in all 0.44°/0.11° domains sea surface temperature (which has a major impact because about 50% of the European model domain is covered by ocean) is deducted from ERA-Interim. Since the 0.44° domains are only slightly larger than the 0.11° domains and hence model internal variability can only slightly

introduce deviations from ERA-Interim within this narrow boundary zone (area of 0.44° domain minus 0.11° domain), the effect of these two nesting strategies is expected to be minor. These nesting issues will be shortly addressed in the discussion.

*RC: [..] Also the conclusion section should be rewritten with a more nuanced interpretation of the results. Is convection permitting modeling really needed if, after bias adjustment, results are no better or only modestly improved compared to coarser resolution simulations? Should multi-model, multi-realization, ensembles be employed or rather one highly tuned simulation? For present climate this might be sufficient but such tuning has well demonstrated shortcomings at climate time scales. Clearly there are substantial challenges remaining before these types of simulations can reliably be used for impacts models. At present the authors fail to acknowledge this and I think somewhat overstate their results. Also, the authors claim that recommendations can be made but then fail to deliver on this promise. What general recommendations, if any, can be made based on this study? Does convection permitting modeling only provide added value over particular areas, for particular cases, particular time scales and particular cases/phenomena? The results here certainly seem to point towards such a limited use or at the very least a need to balance expectations with current capabilities.*

AR: We agree that *"there are substantial challenges remaining before these types of simulations can reliably be used for impacts models".* We will acknowledge these. We were a bit too optimistic in the conclusion because we didn't expect the good CCLM 3km results (uncorrected); also when looking at event scale. Large events were simulated plausible (magnitude and dynamics), particularly events induced by large scale frontal systems. **OQ**: difficulties seem to occur under weak synoptic forcing. This can be partly explained with model-internal-variability as it was published first by Kida et al. (1991). Since our nesting strategy does not make use of any nudging technique, the interior grid cells of the RCMs' domains decouple from the synoptic situation as it prescribed by ERA-Interim. This decoupling effect is largest when synoptic forcing is weak and even small deviations in mesoscale dynamics (that are responsible for moisture supply) can have a large impact on the resultant precipitation on a given location. A paper from the CORDEX-FPS community that points out this issue is currently under review (Coppola et al.). This event type issue was discussed in the paper by Fig. 9 and 12. Of course, this raises questions about the usage of bias correction methods that are not aware of temporal and/or spatial displacements of single events as well as future long-term climate projections (ensemble recommended, testing against historical flood data necessary, etc.). We will acknowledge this and will try to give some concluding remarks.

In general, we will add a detailed discussion section, either within the synthesis section or by a separate section, where we will include all the discussion text hidden in the different chapters and extended by the very valuable issues addressed by the reviewers.

Coppola, E., S. Sobolowski, E. Pichelli, F. Raffaele, B. Ahrens , I. Anders, N. Ban, S. Bastin, M. Belda, D. Belusic, A. Caldas-Alvarez, R. M. Cardoso, S. Davolio, A. Dobler, J. Fernandez, L. Fita Borrell, Q. Fumiere, F. Giorgi, K. Goergen, I. Güttler, T. Halenka, D. Heinzeller, Ø. Hodnebrog, D. Jacob, S. Kartsios, E. Katragkou, E. Kendon, S. Khodayar, H. Kunstmann, S. Knist, A. Lavín-Gullón, P. Lind, T. Lorenz, D. Maraun, L. Marelle, E. van Meijgaard, J. Milovac, G. Myhre, H.-J. Panitz, M. Piazza, M. Raffa, T. Raub, B. Rockel, C. Schär, K. Sieck, P. M. M. Soares, S. Somot, L. Srnec, P. Stocchi, M. H. Tölle, H. Truhetz, R. Vautard, H. de Vries, K. Warrach-Sagi, A first-of-its-kind multi-model convection permitting ensemble for investigating convective phenomena over Europe and the Mediterranean, Climate Dynamics, under review.

Kida, H., Koide, T., Sasaki, H. and Chiba, M.: A New Approach for Coupling a Limited Area Model to a Gcm for Regional Climate Simulations, J. Meteorol. Soc. Jpn., 69(6), 723–728, 1991.

*Reviewer #3 specific comments:*

*P3L7-8: The community is well beyond "first attempts". WRF Hydro (a fully coupled distributed hydrological model within WRF) is far enough in its development to be the core model for the United States' National Water Model. https://ral.ucar.edu/projects/wrf_hydro/overview. Other such systems in operation are TerrSysMP which features a 3-D groundwater model coupled to COSMO-CLM (e.g. Keune et al., 2017). Note that I do not make any on their reliability over climate time scales (i.e. simulations around a decade or more).*

AR: We agree, that the idea to couple hydrological models with RCMs is not new. However, our hydrological model KAMPUS is – thanks to the calibration process that is described in this paper - operationally used for flood-forecasts in the Styrian region (see Fig. 1). Driving KAMPUS with RCM-output to test its applicability for climate impact studies in future activities was a dedicated goal of the underlying research project (CHC-FloodS, see Acknowledgements) that funded this paper. Using a fully coupled modelling system, like WRF-Hydro, has many advantages concerning physical consistency etc., but it also has the drawback, that it is limited to the usage of one RCM (WRF in the case of WRF Hydro). In climate impact application ensembles of RCMs need to be used, if uncertainty in projected climate changes should not be underestimated. Hence, in our case it was important to build up CCLM-KAMPUS and WRF-KAMPUS modelling chains and in this aspect this is the first attempt. We will correct this in the paper. By the way: TerrSysMP is a groundwater model that makes use of a surface runoff module that is also part of WRF Hydro, KAMPUS is a specified flood model calibrated to our catchments.

*P3L13: Please be clear that when you write "coupled" you mean limited one-way coupling (I actually wouldn't call this coupling at all) wherein there is no feedback between the hydrological model and the atmospheric model and the atmospheric model only passes temperature and precipitation.*

AR: We will clarify that: It is a sequential coupling with no feedback.

*P4 L1-7: What are the potential ranges of observational uncertainty? In addition to sensor errors and under catch there is also uncertainty resulting from interpolating to a grid from point based station data. How are these taken into account?*

AR: The hydrological model calibration aims to remove systematic input errors. Of course, other interpolation methods would yield different model parameters. But accuracy depends mainly on the number of stations and available additional information, not so much on the interpolation method (see work of U. Haberlandt et al.). To analyse this would be beyond of this paper. We used all available station data in the area with the dense network of daily recording rain gauges as additional information. Snow under catch is accounted for by a model parameter (snow correction factor) which is calibrated.

*P4L8: The version of WRF used here is quite old (almost 8 years!) and many of the issues related to this version have been corrected. In fact WRF is now 6 full versions more advanced as of this writing. How might this have affected to the results?*

AR: WRF 3.3.1 is the main model version that has been extensively used in the regional climate modelling initiative EURO-CORDEX. The 0.44° and 0.11° simulations have been conducted by Klaus Görgen and contributed to EURO-CORDEX. In our 0.03° simulation we did not want to divert too much from the EURO-CORDEX version for comparability reasons. We agree that the model was improved multiple times since then. One of the major improvements was developed based on our 3.3.1 simulation: we found a bug in the treatment of lateral boundary conditions in the original 3.3.1 version that introduced unphysical artefacts and caused unforeseeable model crashes after simulating two years. The fixed code entered the official release in version 3.7 (see http://www2.mmm.ucar.edu/wrf/users/wrfv3.7/updates-3.7.html topic: "improved specified bdy for long simulations"). In our 3.3.1 version this bug is already fixed. We agree that there might be other changes in the code that would significantly change the results. We will acknowledge this in the conclusion section accordingly.

*P6-Error Correction: More details on the bias adjustment are needed given that it is being pitched as a "novel" approach. How does it perform relative to other approaches? What are its limitations and/or tradeoffs? What are the implications of univariate approach to bias adjustments when the two variables corrected, temperature and precipitation, are related to each other? Also more explanation of the issues/limitations behind current approaches is needed. Maraun et al (2017) have an excellent overview of the current state of bias adjustment shortcomings and placing the SDM approach among these would be helpful to readers.*

AR: A more detailed description of the bias correction and its effects on the flood modelling will be given (see response to reviewers #1 and #2).

*P8L29: "relatively".*

AR: OK

*P12L1: Specify which figure/panel you are referring to.*

AR: OK

*P13L10: I don't believe it has been demonstrated that CPM is "absolutely necessary". I would suggest either making a stronger argument with stronger supporting evidence or modify this claim.*

AR: OK, see response to general comments above and previous reviewers.

*P17L23: The authors write that "performance using 3km data is still best" but it is hard to discern this from the figures. Figure 11 show 24 seasons in total and of those the 3km is closest to observations in only 11 of these. In the other seasons either the 0.11 or 0.44 degrees simulation is closer to observations or the performance across resolutions is equal.*

AR: Indeed, results of all the different RCMs after bias correction are similar (deviations are - with some exceptions - only between 1 or 2 events), which can be interpreted as accurate.

*P16L10: Performance after bias adjustment is degraded over the smallest catchment. Yet this is precisely the type of application (i.e. small catchments) where the authors argue we see the greatest added value of convection permitting modeling. Here it appears that the two techniques, high-resolution modeling and bias adjustment, are not working in concert but in opposition.*

AR: This is true and we will address this. We analysed the effect of the 3hr aggregation (see response above – p.5, bottom)

*P20L30-31: This is in direct contradiction to earlier, and later statements, that CP scales are "absolutely necessary". I would say rather that it is clear that there is still quite some work to do before these models can be reliably used for these sorts of applications.*

AR: We agree.

*P24L10: See previous comment. I do not think the authors have presented evidence sufficient to make this statement.*

AR: We agree. Essential is too strong. But the results using CCLM before bias correction are really promising (and surprisingly good at the same time)

*P25L17-18: Clearly CCLM has higher performance than WRF in this study. However, the authors never discussed the nesting strategy (see table 2), which is different for each model system. Specifically, WRF goes through an additional intermediate nest, a step that will certainly have an impact. How then are then are the WRF and CCLM simulations directly comparable?*

AR: Both nesting strategies (see above) are frequently used in regional climate modelling frameworks. From the climate modelling point of view, both strategies are equally binned. The goal in our study was to investigate how well statistical properties of floods can be simulated by physically based modelling chains in usual climate modelling frameworks. Having this in mind, the nesting strategy becomes only relevant if unphysical perturbations are introduced via one or the other nesting technique, which is not the case. Nevertheless, we will make more statements about the nesting strategy.

*P25L4: What "recommendations", specifically, can be made?*

AR: We agree, due to the unsystematic results and the small test domain general recommendations are hard to give, we will add some specific recommendations - or better remarks - regarding the results of the paper, open questions and possible solutions (e.g., CCLM 3km results promising, but trade-off with computational costs, seasonality important – indication for an accurate representation of atmospheric processes, error correction degrades results – should error correction compensate large errors?, test with historical data necessary).

*P25L5: The modeling systems used here are not "coupled" they are used in a model chain*

AR: Sequential coupling. We will clarify this.

*Figure 1. It is almost impossible to make out the catchment boundaries and initials in this busy figure. I suggest moving the catchment labeling to the larger figure 3.*

*Figure 3. Place catchment labels here in bold. Also bold lines around the catchments themselves so that the six catchments under investigation are clearly delineated.*

AR: Fig. 1 and Fig 3 will be re-structured. OK, labels will be moved to Fig. 3.

*Figure 4. What region is shown here?*

AR: Fig. 4 will be removed (see response to reviewer #1, p10)

*Figure 5. Remove the empty panel in the lower right corner.*

AR: OK.

*Figure 7. It is very hard to distinguish between the blue and black circles. Also including red and green is not colorblind friendly. I suggest a different color scale that has greater separation. This comment applies to Figures 7-12 and 14.*

AR: Colours depend also on the printing. We will check colours before final print.